# Non-Cartesian 3D-SPARKLING vs Cartesian 3D-EPI encoding schemes for functional Magnetic Resonance Imaging at 7 Tesla

**Zaineb Amor**[1], **Philippe Ciuciu**[1,2], **Chaithya G. R.**[1,2], **Guillaume Daval-Frérot**[1,2,3], **Franck Mauconduit**[1], **Bertrand Thirion**[1,2], **Alexandre Vignaud**[1]

**1** CEA, Joliot, NeuroSpin, Université Paris-Saclay, Gif-sur-Yvette, France, **2** Inria, MIND team, Université Paris-Saclay, Palaiseau, France, **3** Siemens Heathineers, Courbevoie, France

* Alexandre.Vignaud@cea.fr

**Data Availability Statement:** The GitHub links of the different python packages used for this work, namely the pysap-mri plugin from the pySAP

## Abstract

The quest for higher spatial and/or temporal resolution in functional MRI (fMRI) while preserving a sufficient temporal signal-to-noise ratio (tSNR) has generated a tremendous amount of methodological contributions in the last decade ranging from Cartesian vs. non-Cartesian readouts, 2D vs. 3D acquisition strategies, parallel imaging and/or compressed sensing (CS) accelerations and simultaneous multi-slice acquisitions to cite a few. In this paper, we investigate the use of a finely tuned version of 3D-SPARKLING. This is a non-Cartesian CS-based acquisition technique for high spatial resolution whole-brain fMRI. We compare it to state-of-the-art Cartesian 3D-EPI during both a retinotopic mapping paradigm and resting-state acquisitions at 1mm$^3$ (isotropic spatial resolution). This study involves six healthy volunteers and both acquisition sequences were run on each individual in a randomly-balanced order across subjects. The performances of both acquisition techniques are compared to each other in regards to tSNR, sensitivity to the BOLD effect and spatial specificity. Our findings reveal that 3D-SPARKLING has a higher tSNR than 3D-EPI, an improved sensitivity to detect the BOLD contrast in the gray matter, and an improved spatial specificity. Compared to 3D-EPI, 3D-SPARKLING yields, on average, 7% more activated voxels in the gray matter relative to the total number of activated voxels.

## Introduction

Functional MRI (fMRI) is currently one of the most commonly used functional neuroimaging techniques to probe brain activity non-invasively through the blood oxygen level-dependent (BOLD) contrast that reflects neurovascular coupling. It offers an interesting trade-off between spatial and temporal resolution in order to study the whole brain as an aggregation of intrinsic functional systems [1]. In particular, by using BOLD fMRI [2] and taking advantage of the neuro-vascular coupling [3], statistical inference about brain activity can be performed at rest [4] and during task performance [5–7]. Whole brain fMRI has been to date, primarily performed at 3 Tesla (3T) at a usual spatial resolution of 2 to 3 mm$^3$ isotropic and a temporal

package for MR image reconstruction and the Nilearn package for fMRI data statistical analysis and visualization, were included within the text of the manuscript. Additional author-generated code can be found in https://github.com/Zaineb18/code_plosone. The clinical protocol approved by the national ethics committee for our experiments and our research center policies prohibits sharing medical data in an open-source fashion. Therefore, we do not publicly share our data. It can, however, be shared with specific organizations and researchers upon special written request at projets.neurospin@cea.fr. This email address targets NeuroSpin's representatives for clinical projects management. Additionally, the corresponding author, Alexandre Vignaud should be contacted at: alexandre.vignaud@cea.fr.

**Funding:** Chaithya G R was supported by the CEA NUMERICS program, which has received funding from the European Union's Horizon 2020 research and innovation program under the Marie Sklodowska-Curie grant agreement No 800945. This work has received financial support from the Leducq Foundation (Large Equipment de Recherche et Plateformes Technologiques program). Finally, this work was granted access to the HPC resources of IDRIS under the allocation 2021-AD011011153 made by GENCI. There was no additional external funding received for this study. The funders had no role in study design, data collection and analysis, decision to publish, or preparation of the manuscript.

**Competing interests:** Guillaume Daval-Frérot was employed by Siemens Healthineers at the time this work was performed. This does not alter our adherence to PLOS ONE policies on sharing data and materials. The other authors have declared that no competing interests exist.

resolution of 2–3s using 2D echo planar imaging (EPI). This has recently changed with significant progress in two areas, namely the rise of ultra-high magnetic fields (UHF, 7T and beyond) and the possibility to accelerate data acquisition. Both improvements offer the opportunity to reach higher spatio-temporal resolution [8–12].

On the one hand, collecting fMRI data at higher spatial (e.g. submillimetric) resolution is instrumental in obtaining an improved specificity in the gray matter, which enables laminar fMRI [13–15] as it limits the contribution of large draining veins to the BOLD effect [16] elicited by task performance. On the other hand, higher temporal resolution in fMRI has allowed the community to revisit the question and relevance of fast detectable components in the BOLD effect [17, 18], which precede the well-known sluggish haemodynamic response peaking 4 to 6 seconds after stimulus presentation [19]. Additionally, short repetition times in resting-state fMRI make it possible to remove aliasing artifacts caused by heartbeat and breathing rate [20, 21] and thus to study a cleaner neuro-vascular coupling.

Importantly, acquiring high-resolution fMRI data without sacrificing the 3D field-of-view (3D FOV) and using conventional MR systems degrades both the signal-to-noise ratio (SNR) [22] and temporal SNR (tSNR) as the two indices are linearly linked in the thermal noise regime [23]. As the tSNR is a proxy of the temporal stability in fMRI, significant efforts have been deployed to mitigate this issue either by moving to stronger magnetic fields (7T and beyond) or by using advanced coil designs or both [24]. Combining high spatial and high temporal resolution can be achieved using more powerful MR gradient systems or accelerated acquisition schemes. As the latter is a more cost-effective and versatile option, significant efforts have been pushed forward to develop and implement more efficient encoding techniques and image reconstruction methods.

Under-sampling the k-space in the Cartesian framework was made possible through the progress of parallel imaging (PI) [25–28] and the corresponding reconstruction methods such as SENSE [29] or GRAPPA [30]. These developments helped improve the spatio-temporal resolution in whole brain fMRI: By using multi-band excitation pulses, 2D SMS-EPI [31, 32] is able to acquire several slices using a single shot. 3D-EPI [33] has the same acceleration capabilities than 2D SMS-EPI as they both enable acceleration in-plane and along the partition encoding direction, however, 3D-EPI requires lower flip angles and provides a better SNR. Nevertheless, 3D-EPI is more sensitive to physiological noise than 2D SMS-EPI. In [34], the authors report very comparable performances between 3D-EPI and 2D SMS-EPI. That being said, the high specific absorption rate (SAR) associated with 2D SMS-EPI strategy constitutes a limitation for in vivo applications at UHF. Furthermore, CAIPIRINHA for multi-slice imaging [35] and 2D CAIPIRINHA [36] for 3D acquisitions can be applied to improve the g-factor map [37, 38] which quantifies, among other things, the effect of the sampling pattern on the noise spatial distribution in the parallel imaging setting.

Despite offering shorter scan times, PI-based accelerated Cartesian strategies remain limited due to the suboptimal gradient waveforms they use. This has motivated the development of non-Cartesian encoding patterns such as radial and spiral as a way to further enhance the sampling efficiency by enabling a denser coverage of the low frequencies and an optimal use of the gradient power. In addition to a faster coverage of the k-space, non-Cartesian strategies yield fewer (but more complex) coherent under-sampling artifacts. This feature, combined with the difficulty of accurately regridding non-Cartesian samples into a Cartesian grid, has led to a shift in reconstruction methods: The non-Uniform fast Fourier Transform (NUFFT) gained ground over the fast Fourier Transform (FFT) and new iterative reconstruction algorithms such as conjugate gradient SENSE (CG SENSE) [39] or SPIRiT [40] have been

implemented. Spiral-based acquisition schemes are today, the most used non-Cartesian alternatives for high-resolution fMRI [41–43].

Even though a more efficient traversal of the k-space was the main motivation behind their development, non-Cartesian methods naturally implement variable density sampling (VDS), thereby implicitly capitalizing on the idea that some data points bring more knowledge than others. However, given the deterministic character of their trajectories, they do not fully exploit an important notion that emerged recently: MR images are inherently sparse or compressible in a well-chosen transform domain like wavelets, for instance.

Following the later idea and in the context of further shortening the acquisition time, compressed sensing (CS) was first introduced in the MRI field [44] in an attempt to reach higher acceleration factors and, therefore, better image resolution without extending the scan time. CS can be optimally combined with multi-coil acquisition [45, 46], and allows us to accurately reconstruct images from highly under-sampled measurements using three complementary ingredients: VDS, locally uniform coverage of k-space and nonlinear sparsity-promoting reconstruction methods. Successful applications of CS-MRI implement VDS where the central portion of k-space is oversampled compared to its periphery [47, 48].

The application of CS to dynamic MR imaging, such as fMRI, should optimally enforce sparsity both in space and time. However, sparsity in the time domain is difficult to achieve in fMRI applications as the recorded fMRI signal is not quasi-periodic in contrast to the heartbeat in cardiac cine imaging, for instance. Hence, finding a sparsifying basis for the BOLD signal, such as activelets [49], has been a research topic in itself. This is why the application of CS in fMRI [50–53] remains sparse and limited. Moreover, the k-space under-sampling is never performed according to the three encoding directions in an isotropic way in 3D.

Recently, the CS-based SPARKLING (Spreading Projection Algorithm for Raping K-space sampLING) encoding scheme was introduced for 2D [54] and 3D [55, 56] $T_2^*$w MR anatomical imaging. 3D-SPARKLING [56] generates non-Cartesian multi-shot pseudo-random sampling patterns that fit a non-uniform target sampling density. These patterns perform VDS and are locally uniform. The shots generation is optimization-driven and ensures that they are fully optimized in 3D. A projection step in the optimization algorithm ensures that each shot fits the hardware constraints (gradient magnitude and slew rate) as well as the contrast constraint (same echo time).

In this work, we demonstrate the potential of 3D-SPARKLING for fMRI for the first time by comparing it with state-of-the-art 3D-EPI at 1mm$^3$ and a temporal resolution (volumetric TR) of 2.4s for whole-brain fMRI at 7T. 3D-SPARKLING was used in a scan-and-repeat mode and assessed during retinotopic mapping in task-based fMRI as well as during resting-state fMRI acquisitions in order to demonstrate its potential as a suitable alternative to 3D-EPI.

## Theory

3D-SPARKLING implementation is detailed in [56] and is based on earlier works [54, 57]. Hereafter, we briefly explain the theory underpinning the trajectories design. More details can be found in [54, 56–58].

Let $\mathbf{K} = (\mathbf{k}_i)_{i=1}^{N_c}$ be a segmented 3D sampling pattern consisting of $N_c$ shots and each shot has $N_s$ sampling points. Each shot can be written in 3D as follows $\mathbf{k}_i(t) = (k_{i,x}(t), k_{i,y}(t), k_{i,z}(t))$. K-space trajectories in MRI are run using varying magnetic field gradients according to the following equation $\mathbf{k}_i(t) = \frac{\gamma}{2\pi} \int_0^t \mathbf{G}_i(\tau) d\tau$ where $\mathbf{G}_i(t) = (G_{i,x}(t), G_{i,y}(t), G_{i,z}(t))$.

Let $K_{max}$ define the boundaries of a 3D k-space domain such as $K_{max} = K^x_{max} = K^y_{max} = K^z_{max}$ for simplicity. 3D-SPARKLING shots are defined according to the following set of constraints:

$$\mathcal{Q}^{N_c}_{\alpha,\beta} = \left\{ \begin{array}{c} \forall i = 1, \ldots, N_c, \quad \mathbf{k}_i \in \mathbb{R}^{3 \times N_s}, \\ \mathbf{A}\mathbf{k}_i = \mathbf{v}, \\ \|\dot{\mathbf{k}}_i\|_{2,\infty} \leq \alpha, \quad \|\ddot{\mathbf{k}}_i\|_{2,\infty} \leq \beta, \end{array} \right\} \tag{1}$$

where $|\mathbf{c}|, \infty = \sup_{0 \leq n \leq N_s-1} (|c_x[n]|^2 + |c_y[n]|^2 + |c_z[n]|^2)^{1/2}$. $\dot{\mathbf{k}}_i$ and $\ddot{\mathbf{k}}_i$ are the first and second-order time derivatives of the shot denoted $\mathbf{k}_i$. $\alpha = \frac{1}{K_{max}} \min\left(\frac{\gamma G_{max}}{2\pi}, \frac{1}{FOV \cdot \delta t}\right)$ and $\beta = \frac{\gamma S_{max}}{2\pi K_{max}}$ with $G_{max}$ and $S_{max}$ the maximum gradient and slew rate constraints respectively.

$\mathbf{A}$ and $\mathbf{v}$ define the linear constraints that ensure that each shot crosses the center of k-space at the echo time (TE).

3D-SPARKLING trajectories are optimized to 1) match a target sampling density $\rho$ and 2) optimally cover the k-space $\Omega = [-1, 1]^3$ in a locally uniform way according to the following formulation. Given a target sampling density $\rho : \Omega \to \mathbb{R}$ defined such as $\rho(x) \geq 0$ for all $x$ and $\int \rho(x)dx = 1$, the sampling pattern $\mathbf{K} \in \Omega^p$ is generated by minimizing the problem defined by:

$$\widehat{\mathbf{K}} = \arg\min_{\mathbf{K} \in \mathcal{Q}^{N_c}_{\alpha,\beta}} F_p(\mathbf{K}) = [F^a_p(\mathbf{K}) - F^r_p(\mathbf{K})] \tag{2}$$

with

$$F^a_p(\mathbf{K}) = \frac{1}{p} \sum_{i=1}^{p} \int_{\Omega} H(x - \mathbf{K}[i]) \rho(x) \, dx$$

and

$$F^r_p(\mathbf{K}) = \frac{1}{2p^2} \sum_{1 \leq i,j \leq p} H(\mathbf{K}[i] - \mathbf{K}[j]) .$$

$F^a_p$ is an attraction term that guarantees 1) while $F^r_p$ is a repulsion term that ensures 2) and $H(x) = \|x\|_2$. In this formulation, $p = N_c \times N_s$ denotes the total number of sampling points in $\mathbf{K}$ and $\mathcal{Q}^{N_c}_{\alpha,\beta}$ denotes the set of hardware and linear constraints defined earlier.

## Methods

### Experimental setup and stimulation paradigm

Data was collected at 7T MRI (7T Siemens Magnetom scanner, Erlangen, Germany) from seven healthy volunteers (3 females, 4 males) aged between 20 and 40 years old with normal-to-corrected vision, and using a 1Tx-32Rx head coil (Nova Medical, Willimgton, CO, USA). Data from only 6 volunteers (1 male excluded) was actually involved in this work as the last volunteer felt asleep during the experiment. The experimental protocol was approved by the national ethics committee (Comité de Protection des Personnes) under the protocol identifier CPP 100048 (CPP Sud Méditerranée 4 number 180913, IDRCB:2018-A01176l–53). All participants gave their written informed consent.

The functional data was collected with $T_2^*$-weighted 3D-EPI and 3D-SPARKLING encoding schemes. Task-based fMRI data was acquired using a classical retinotopic mapping experimental paradigm with a 32s-period rotating wedge (2 consecutive runs per sequence: clockwise and counter-clockwise). In what follows, we denote this rotation as $P_{cyc} = 32$. The code of the stimulation can be found in [59] The choice of the retinotopic mapping paradigm emanated

| | **Resting-State** | | **Retinotopic Mapping** | | | |
|---|---|---|---|---|---|---|
| | 5 minutes | 5 minutes | 4 minutes 48 seconds | 4 minutes 48 seconds | 4 minutes 48 seconds | 4 minutes 48 seconds |
| V#1 | | | *3D-SPARKLING Clockwise* | *3D-SPARKLING Counter-clockwise* | *3D-EPI Clockwise* | *3D-EPI Counter-clockwise* |
| V#2 | *3D-EPI* | *3D-SPARKLING* | *3D-EPI Clockwise* | *3D-EPI Counter-clockwise* | *3D-SPARKLING Clockwise* | *3D-SPARKLING Counter-clockwise* |
| V#3 | *3D-SPARKLING* | *3D-EPI* | *3D-SPARKLING Clockwise* | *3D-SPARKLING Counter-clockwise* | *3D-EPI Clockwise* | *3D-EPI Counter-clockwise* |
| V#4 | *3D-EPI* | *3D-SPARKLING* | *3D-EPI Clockwise* | *3D-EPI Counter-clockwise* | *3D-SPARKLING Clockwise* | *3D-SPARKLING Counter-clockwise* |
| V#5 | *3D-SPARKLING* | *3D-EPI* | *3D-SPARKLING Clockwise* | *3D-SPARKLING Counter-clockwise* | *3D-EPI Clockwise* | *3D-EPI Counter-clockwise* |
| V#6 | *3D-EPI* | *3D-SPARKLING* | *3D-EPI Clockwise* | *3D-EPI Counter-clockwise* | *3D-SPARKLING Clockwise* | *3D-SPARKLING Counter-clockwise* |

**Fig 1. Time course of the resting-state and task-based fMRI sessions.** From subject V#2 to V#6, rs-fMRI data was systematically collected prior to retinotopic mapping data to avoid any contamination in the visual network due to task performance. The order in which 3D-EPI and 3D-SPARKLING were run was randomized across individuals.

from two main factors: First, a higher tSNR available in the visual cortex and second, the fact that passive viewing is less prone to variability or errors in task performance [60]. For technical reasons, resting-state fMRI data was collected for only five volunteers out of the six in order to evaluate the tSNR. The order in which 3D-EPI and 3D-SPARKLING sequences were run was randomized across individuals. Resting-state data was always collected before task-based fMRI. Fig 1 illustrates how the full fMRI experiment was conducted in each participant.

## Data acquisition and sequence parameters

Functional data was collected at a 1-mm isotropic and 2.4s spatio-temporal resolution. The same sequence parameters were used for 3D-SPARKLING and 3D-EPI as shown in Table 1. 3D-EPI readouts were accelerated along the phase encoding direction and the partition encoding direction by a factor of 4 and 2, respectively. Partial Fourier (6/8) was also applied along both phase encoding directions. CAIPIRINHA 2D [36] reconstruction was applied to the 3D-EPI data.

**Table 1. Specifications of the different pulse sequences: 3D-SPARKLING and 3D-EPI were used to acquire the fMRI data at 1mm³ and 2.4s-$TR_v$ (volumetric TR) resolution. The native orientation was kept the same for the acquisitions associated with both sequences: Oblique transverse orientation. The GRE sequence used for external field maps had 3 echoes. An additional single-repetition 3D-EPI in A-P encoding was acquired for $\Delta B_0$ correction on 3D-EPI. The MP2RAGE sequence was used to acquire an anatomical $T_1$w scan.**

| | 3D-SPARKLING | 3D-EPI | 3D GRE | 3D-EPI | MP2RAGE |
|---|---|---|---|---|---|
| TE(ms) | 20 | 20 | 1.8, 3.06, 5.10 | 20 | 3.29 |
| Unitary TR(ms) | 50 | 50 | 20 | 50 | 5000 |
| Volumetric TR(s) | 2.4 | 2.4 | 58 | 2.4 | 347 |
| Spatial resolution (mm) | 1 | 1 | 3 | 1 | 1 |
| FOV(mm) | (192,192,128) | (192,192,128) | (192,192,132) | (192,192,128) | (192,192,128) |
| Number of repetitions | 120 125 | 120 125 | 1 | 1 | 1 |
| Encoding direction | | P-A | P-A | A-P | R-L |

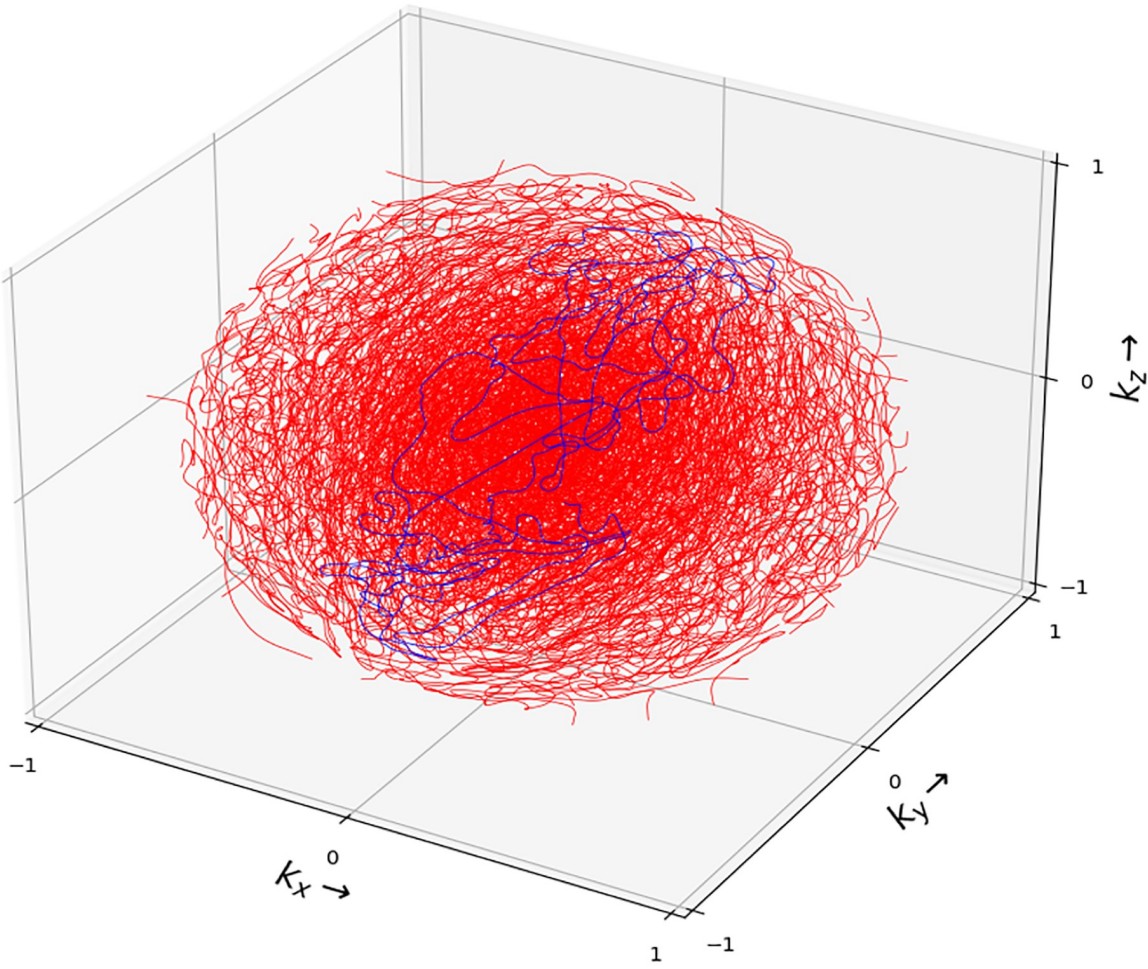

**Fig 2. 3D-SPARKLING sampling scheme.** The sampling scheme is segmented into 48 readouts plotted in red except for one plotted in blue for the sake of visualization.

3D-SPARKLING encoding pattern consisted of 48 shots generated to match the target sampling density that yielded the trajectories (Fig 2) with the best point spread function (PSF) in terms of full width at half maximum (FWHM) and maximum-to-side lobe ratio. The best identified cutoff $C$ and decay $D$ parameters of the target density $\pi_{C,D}$ were set respectively to $C = 0.25$ and $D = 2$ (Eq (3)) to separate out the sampling regimes in the low and high frequencies as follows:

$$\pi_{C,D}(x) = \begin{cases} \kappa & |x| < C \\ \kappa \left(\dfrac{C}{|x|}\right)^{D} & |x| > C \end{cases} \tag{3}$$

where $\kappa$ is a normalizing constant.

Both readouts had very close observation times: 26.33ms for 3D-EPI and 26.88ms for 3D-SPARKLING and both were acquired in the P-A encoding direction. Additionally, the two 3D gradient recalled echo (GRE) sequences involved a gradient spoiling as well as a RF spoiling.

A 15s-calibration sequence was run at the beginning of each 3D-EPI sequence (be it single or multi-repetition) to estimate coil sensitivity maps for 3D-EPI data and at the same time ensure steady-state. An additional single-repetition 3D-EPI acquisition in the opposite A-P encoding direction was then performed to correct 3D-EPI functional data for $\Delta B_0$ distortions using the so-called TOPUP approach [61, 62]. In contrast, for 3D-SPARKLING non-Cartesian data, we used a distinct 3D GRE sequence with three consecutive echoes to estimate both an external field map ($\Delta B_0$) and external coil sensitivity maps. In order to have a reasonable acquisition time, the just-mentioned sequence was run for a spatial resolution of 3mm$^3$ as reported in Table 1. Raw data of the first echo from the GRE sequence was used to compute the sensitivity maps whereas the 3 echoes were used to obtain an accurate estimate of the $\Delta B_0$ field map. The latter was extrapolated in the image domain to fit the spatial resolution and 3D FOV of the 3D-SPARKLING fMRI scans. The sensitivity maps were extrapolated in the k-space domain for the same reason. Steady-state was achieved in 3D-SPARKLING acquisitions by means of dummy scans corresponding to 960 unitary repetition times.

## fMR image reconstruction and preprocessing

The 3D-EPI volumes were reconstructed independently from each other using a calibrated multi-coil reconstruction which involves the 15s-calibration sensitivity maps mentioned in the section above. The reconstruction is implemented using the vendor's ICE software. Since navigators were implemented into the 3D-EPI sequence, we corrected the 3D-EPI volumes for zeroth order dynamic $\Delta B_0$ fluctuations. 3D-EPI data was corrected for off-resonance artifacts using the additional single-repetition A-P acquisition, and the TOPUP approach available in FSL [63].

The 3D-SPARKLING volumes were reconstructed independently from each other as well as using a calibrated multi-coil CS-based reconstruction method [64] and the external sensitivity maps mentioned earlier. This reconstruction method involves a sparsity promoting prior ($\ell_1$-norm) in the wavelet domain and performs the optimization using the proximal optimized gradient method (POGM) algorithm [65]. This approach is implemented in the pysap-mri [66] plugin [67] of the pySAP package [68]. The regularization parameter of this reconstruction was carefully chosen following a line-search and assessing its impact on image quality visually as well as on the temporal aspect of the rs-fMRI data (tSNR and motion regression). Additionally, static $\Delta B_0$ was corrected during single volume MR image reconstruction using the external $\Delta B_0$ map mentioned earlier. The corresponding frequency offset $\delta\omega_0 = \gamma\Delta B_0$ was actually injected in the definition of the non-Fourier forward operator $\tilde{\mathbf{F}}_\Omega$ that is classically used to refine the MR signal model (see for instance [69, Eq. (2)]). The complete reconstruction and correction pipeline we used for each 3D-SPARKLING volume is the same as that described in [69, Supp. Mat.]. Its open-source implementation, which follows the method proposed in [70], is available in the pysap-mri package as well.

Motion correction was applied similarly to 3D-EPI and 3D-SPARKLING fMRI data using SPM12 [71]. Similarly, co-registration of the functional and the anatomical (i.e. $T_1$-weighted) volumes was performed using SPM12. Except for estimating BOLD phase maps (cf. Section "Accuracy of the retinotopic phase maps"), no spatial smoothing was applied to the volumes in order to keep the advantages of the native 1-mm isotropic spatial resolution.

## Design matrix for capturing task-related BOLD signal fluctuations

For each participant, the sequence of task-related fMRI volumes was analyzed using a two-session first-level general linear model (GLM) that is summarized through a design matrix $\mathbf{X} \in \mathbb{R}^{2N_{\text{vol}} \times Q}$ with $N_{\text{vol}} = 120$ scans per run and $Q$ defines the number of regressors as detailed

hereafter. The objective was to estimate the parameters of interest $\boldsymbol{\beta} = (\beta_i)_{i=1}^{V} \in \mathbb{R}^{Q \times V}$ in $V$ voxels from the observed BOLD fMRI time series $\mathbf{Y} = (\mathbf{y}_i)_{i=1}^{V} \in \mathbb{R}^{2N_{\mathrm{vol}} \times V}$, in the classical massively univariate linear model: $\mathbf{Y} = \mathbf{X}\boldsymbol{\beta} + \mathbf{N}$, where $\mathbf{N} = (\mathbf{n}_i)_{i=1}^{V} \in \mathbb{R}^{2N_{\mathrm{vol}} \times V}$ stands for the voxel-wise additive serially correlated Gaussian noise term.

Due to the repetition of the task over two consecutive sessions, matrix $\mathbf{X}$ has block-diagonal structure:

$$\mathbf{X} = \begin{pmatrix} \mathbf{X}_1 & \mathbf{0}_{N_{\mathrm{vol}},Q/2} \\ \mathbf{0}_{N_{\mathrm{vol}},Q/2} & \mathbf{X}_2 \end{pmatrix},$$

where the non-zero diagonal blocks $\mathbf{X}_1$ and $\mathbf{X}_2$ are respectively associated with the experimental paradigm that is carried out during the first and second sessions, namely the clockwise and counter-clockwise rotating wedges. Each block $\mathbf{X}_s$ is composed of $Q/2 = 10$ regressors defined as follows:

$$\mathbf{X}_s = \begin{pmatrix} \mathbf{x}_{s,1}^{\mathrm{task}} & \mathbf{x}_{s,2}^{\mathrm{task}} & \mathbf{x}_{s,3}^{\mathrm{mot}} & \dots & \mathbf{x}_{s,8}^{\mathrm{mot}} & \mathbf{x}_{s,9}^{\mathrm{pol}} & \mathbf{x}_{s,10}^{\mathrm{bas}} \end{pmatrix} \in \mathbb{R}^{N_{\mathrm{vol}} \times Q/2}, \tag{4}$$

where two paradigm-related parametric and continuous regressors $\mathbf{x}_{s,1}$ and $\mathbf{x}_{s,2}$ serve to capturing the BOLD fluctuations elicited by the stimulus presentation. For the clockwise session ($s = 1$), we used

$$\begin{cases} x_{1,1}^{\mathrm{task}}(t) &= \cos\left(-\dfrac{\pi}{2} - \dfrac{2n_{\mathrm{cyc}} \, t \, \pi}{N_{\mathrm{vol}}}\right) = -\sin\left(\dfrac{2n_{\mathrm{cyc}} \, t \, \pi}{N_{\mathrm{vol}}}\right) \\[3mm] x_{1,2}^{\mathrm{task}}(t) &= \sin\left(-\dfrac{\pi}{2} - \dfrac{2n_{\mathrm{cyc}} \, t \, \pi}{N_{\mathrm{vol}}}\right) = -\cos\left(\dfrac{2n_{\mathrm{cyc}} \, t \, \pi}{N_{\mathrm{vol}}}\right) \end{cases} \tag{5}$$

where $n_{\mathrm{cyc}} = N_{\mathrm{vol}} \times \mathrm{TR}/P_{\mathrm{cyc}} = 9$, while for the counter-clockwise session ($s = 2$) we define the first regressor turning in the opposite direction:

$$\begin{cases} x_{2,1}^{\mathrm{task}}(t) &= \cos\left(-\dfrac{\pi}{2} + \dfrac{2n_{\mathrm{cyc}} \, t \, \pi}{N_{\mathrm{vol}}}\right) = -x_{1,1}^{\mathrm{task}}(t) \\[3mm] x_{2,2}^{\mathrm{task}}(t) &= \sin\left(-\dfrac{\pi}{2} + \dfrac{2n_{\mathrm{cyc}} \, t \, \pi}{N_{\mathrm{vol}}}\right) = x_{1,2}^{\mathrm{task}}(t) \end{cases}. \tag{6}$$

Next, 6 session-specific motion-related regressors $(\mathbf{x}_{s,3}^{\mathrm{mot}}, \dots, \mathbf{x}_{s,8}^{\mathrm{mot}})$, three for rotations and three for translations, were estimated using `SPM12` while a simple polynomial regressor $x_{s,9}^{\mathrm{pol}}(t) = t$ was fitted to capture the linear trend on top of the mean signal modeled by $x_{s,10}^{\mathrm{bas}}(t) = 1, \forall t$.

The parameter estimates $\hat{\boldsymbol{\beta}} = (\hat{\beta}_i)_{i=1}^{V}$ were estimated in the maximum likelihood sense using the `Nilearn` [72] package which implements a prewhitening procedure based on a first-order autoregressive noise model for $\mathbf{N}$. Massive univariate analysis was thus carried out to obtain parameter estimates $\hat{\boldsymbol{\beta}}$ within the brain mask composed of $V$ voxels where $V$ varies between 1,449 299 and 1, 649 380 across participants (approximately one fourth of the 3D FOV). For the sake of simplicity, the brain mask was computed from the mean volume of the 3D-EPI scan sequence using `compute_epi_mask` from `Nilearn`.

## Statistical analysis of the retinotopic mapping data

Firstly, a Fisher-test was constructed to estimate the global effect of interest associated with the task-related regressors, namely the reduced model encoded by
$\mathbf{X}_0 = (\mathbf{x}_{1,1}^{\text{task}}, \mathbf{x}_{1,2}^{\text{task}}, \mathbf{x}_{2,1}^{\text{task}}, \mathbf{x}_{2,2}^{\text{task}}) \in \mathbb{R}^{2N_{\text{vol}} \times 4}$. The corresponding null hypothesis reads $H_{0,EOI} : \mathbf{C}^T \boldsymbol{\beta} = 0$ where $\mathbf{C}^T \in \{0, 1\}^{4 \times Q}$ is a contrast matrix given by:

$$
\mathbf{C}^T = \begin{pmatrix}
\overbrace{1}^{\mathbf{x}_{1,1}^{\text{task}}} & \overbrace{0}^{\mathbf{x}_{1,2}^{\text{task}}} & \dots & \dots & \overbrace{0}^{\mathbf{x}_{2,1}^{\text{task}}} & \overbrace{0}^{\mathbf{x}_{2,2}^{\text{task}}} & \dots & \dots & 0 \\
0 & 1 & 0 & \dots & 0 & 0 & \dots & \dots & 0 \\
0 & 0 & \dots & \dots & 1 & 0 & \dots & \dots & 0 \\
0 & 0 & \dots & \dots & 0 & 1 & 0 & \dots & 0
\end{pmatrix}. \tag{7}
$$

The degrees of freedom of the $F_{v_1, v_2}$-test are given by $v_1 = rank(\mathbf{X}) - rank(\mathbf{X}_0) = 16$ and $v_2 = 2N_{\text{vol}} - rank(\mathbf{X}) = 220$. In practice, $\hat{\boldsymbol{\beta}}$ was used in the computation of the residual sums of squares:

$$
\text{RSS} = \|\mathbf{Y} - \mathbf{X}\hat{\boldsymbol{\beta}}\|_2^2 \quad \text{and} \quad \text{RSS}_0 = \|\mathbf{Y} - \mathbf{X}_0 \hat{\boldsymbol{\beta}}_0\|_2^2 \tag{8}
$$

where $\hat{\boldsymbol{\beta}}_0 = (\hat{\beta}_{0,i})_{i=1}^V$ with $\hat{\beta}_{0,i} = (\hat{\beta}_{1,i}, \hat{\beta}_{2,i}, \hat{\beta}_{11,i}, \hat{\beta}_{12,i})^T$.

These results were then used to form the Fisher statistics $F_{v_1, v_2} = \frac{\text{RSS}_0 - \text{RSS}}{\text{RSS}}$. The resulting whole brain statistical $F_{v_1, v_2}$-map was thresholded according to three different strategies where the null hypothesis $H_{0,EOI}$ was rejected if the p-value $p$ met at least one of the following criteria:

(i): $p < 0.05$ with false discovery rate (FDR) correction [73];

(ii): $p < 0.001$ without correcting for multiple comparisons;

(iii): $p < 0.05$ without correcting for multiple comparisons and with a minimum cluster size of 5 voxels.

In addition to strategies (i) and (ii) which were applied to all participants, alternative (iii) has also been used in participant V#5 for whom the global effect of interest was weak and the tSNR low. In addition, given the high spatial resolution and the whole brain analysis, the false discovery rate correction was too restrictive for the majority of the volunteers.

The individual $F$-maps have been used as statistical masks for further analysis of retinotopic mapping at the subject level as detailed hereafter.

Secondly, in order to derive the retinotopic maps we also formed the Student-$t$ tests that are based on the following elementary null hypotheses:

$$
\begin{cases}
H_{0,1} : & \beta_{1,i} = 0 \quad \forall i = 1 : V, \\
H_{0,2} : & \beta_{2,i} = 0 \quad \forall i = 1 : V, \\
H_{0,3} : & \beta_{11,i} = 0 \quad \forall i = 1 : V, \\
H_{0,4} : & \beta_{12,i} = 0 \quad \forall i = 1 : V.
\end{cases}
$$

We computed the corresponding z-score maps $\mathbf{z} = (z_{j,i})_{j=1:4, i=1:V}$ from which we formed the

voxelwise estimates of the signal phase in a session-specific manner as follows:

$$\begin{cases} \phi_{Clock,i} & = & \arctan\left(\dfrac{-z_{1,i}}{-z_{2,i}}\right) & \forall i = 1:V, \\[2ex] \phi_{CClock,i} & = & \arctan\left(\dfrac{-z_{3,i}}{z_{4,i}}\right) & \forall i = 1:V, \end{cases}$$

where $\phi_{Clock,i}$ and $\phi_{CClock,i}$ respectively stand for the phase estimates associated with the clockwise and counter-clockwise sessions. Then, after compensating for the recorded BOLD response delay ($d_{h,i} = \frac{\phi_{Clock,i} + \phi_{CClock,i}}{2} \quad \forall i = 1:V$) due to the haemodynamic delay in $\phi_{Clock,i}$ and $\phi_{CClock,i} \forall i = 1:V$, we derived the overall retinotopic phase estimate as follows:

$$\phi_i = \frac{\phi_{Clock,i} - \phi_{CClock,i}}{2} \quad \forall i = 1:V. \tag{9}$$

## Metrics used for quality assessment

Resting-state fMRI mean images (i.e. volumes) were visually inspected on an individual basis to evaluate image quality, while the whole scan sequences were used to compute the tSNR metric allowing us to make a comparison between the 3D-EPI and 3D-SPARKLING encoding schemes for each subject.

Complementary to that, retinotopic mapping fMRI data was used to conduct both qualitative and quantitative assessment at the subject level through a series of metrics respectively referenced to as q- and Q-metric in the following in order to compare the statistical performances of 3D-EPI and 3D-SPARKLING in terms of sensitivity and specificity. First, the consistency of activation maps between the two acquisition techniques was evaluated according to the following qualitative (q) and quantitative (Q) criteria:

1) `qCons1`: **Consistency of activation maps**. The z-score maps derived from the global effects of interest (cf. $H_{0,EOI}$) were visually assessed and compared subject-wise.

2) `QCons1`: **Spatial overlap of the statistically significant activation maps**. Binarized activation masks were first generated from the respective 3D-EPI and 3D-SPARKLING above mentioned z-score maps, with z-scores higher (respectively, lower) than 3.09 (corresponding to $p = 0.001$) set to the value of 1 (respectively, 0). Then the Dice index [74] or F1-score was used to measure the overlap between these 3D-EPI and 3D-SPARKLING activation masks, a value closer to 1 (respectively, 0) indicating a strong (respectively, weak) overlap or consistency between activation patterns.

Second, an evaluation of the sensitivity to the BOLD contrast was conducted on the basis of complementary criteria:

3) `QSens1`: **Number of activated voxels overall and in the gray matter**. These figures were computed from the global effect of interest statistical maps (z-scores) after applying a threshold at $p < 0.001$ without multiple comparisons correction in both 3D-EPI and 3D-SPARKLING data sets and compared one another in each participant.

4) `qSens1`: **Significance of the activation patterns**. The activation patterns derived from the data collected in two volunteers (V#3 and V#4) were displayed on the same slices and visually compared. These two volunteers were selected to specifically showcase the session order effect, namely what is the impact of running 3D-SPARKLING or 3D-EPI prior to the other sequence respectively.

5) `QSens2`: **Statistical comparison of the distributions of the significant effects of interest**. A Kolmogorov-Smirnov (KS) test [75] was performed between the distributions of the statistically significant $z$-scores located in the gray matter and associated either with 3D-EPI or 3D-SPARKLING retinotopic fMRI data and denoted respectively $P_{EPI}(z)$ and $P_{SPARK}(z)$. The goal was to assess what distribution had a heavier tail (i.e. was more shifted to the right).

Third, an evaluation of the spatial specificity was carried out according to the following metric:

6) `QSpe1`: **Percentage of activations in the brain tissues**. For each participant, we compared the percentages of activated voxels in gray and white matter (GM and WM, respectively) as well as in the cerebrospinal fluid (CSF) for both encoding schemes after carrying out the tissue segmentation on the anatomical $T_1$-weighted image and the co-registration of the mean fMRI images with the anatomical scan using `SPM12`. This software actually yielded the tissue probability masks for gray and white matter and the CSF.

7) `QSpe2`: **Prevalence of true positives vs false positives**. The total number of activated voxels retrieved after thresholding the $z$-scores at $p < 0.05$ with FDR correction is reported for all participants and compared with the figures corresponding to `QSens1`.

Finally, the accuracy of the BOLD phase maps was assessed qualitatively according to the following criterion:

8) `qAccu1`: **Accuracy of retinotopic mapping**. The volumetric statistical phase maps $\phi = (\phi_i)_{i=1:V}$ (cf. Eq (9)) and their projection onto the pial cortical surface were visually assessed for two volunteers (V#3 and V#4). The projection was performed using `vol_to_surf` from `Nilearn` and the meshes corresponding to the pial, inflated, sulcus and white matter surfaces were computed using `FreeSurfer 7`.

The order of these metrics matters. It has been carefully chosen to progressively demonstrate the benefits associated with 3D-SPARKLING. For the quantitative criteria denoted `QCons1`, `QSens1`, `QSens2`, `QSpe1`, the statistically significant activations were defined by thresholding the $z$-scores at 3.09 (corresponding to a p-value of 0.001).

## Results

### Image quality and temporal SNR

Fig 3 demonstrates on an individual basis the superior image quality yielded by 3D-EPI: Fine-grained anatomical details are lost in 3D-SPARKLING mean fMRI images. Moreover, the between-tissue contrast of 3D-EPI images is clearer compared to that of the 3D-SPARKLING. The two encoding schemes are actually differently affected by the signal loss: 3D-SPARKLING appears less impacted in the frontal lobes (blue arrows) but more severely degraded around the ventricles (orange arrows). Geometric distortions are also differently influencing the 3D-EPI and 3D-SPARKLING images and are mostly visible in the sagittal views. These discrepancies arise from the well-known differences between Cartesian and non-Cartesian imaging in terms of robustness to static and dynamic $B_0$ field inhomogeneities which affect their point spread function (PSF) and therefore their effective spatial resolution differently. Consequently, even after accounting for the impact of the partial Fourier (6/8) applied to 3D-EPI on the PSF, the effective resolution of 3D-SPARKLING fMRI images is lower than that of 3D-EPI images.

The temporal stability of the fMRI signal is usually evaluated through the tSNR. Fig 4 shows tSNR maps derived from 3D-EPI and 3D-SPARKLING resting-state fMRI data and collected

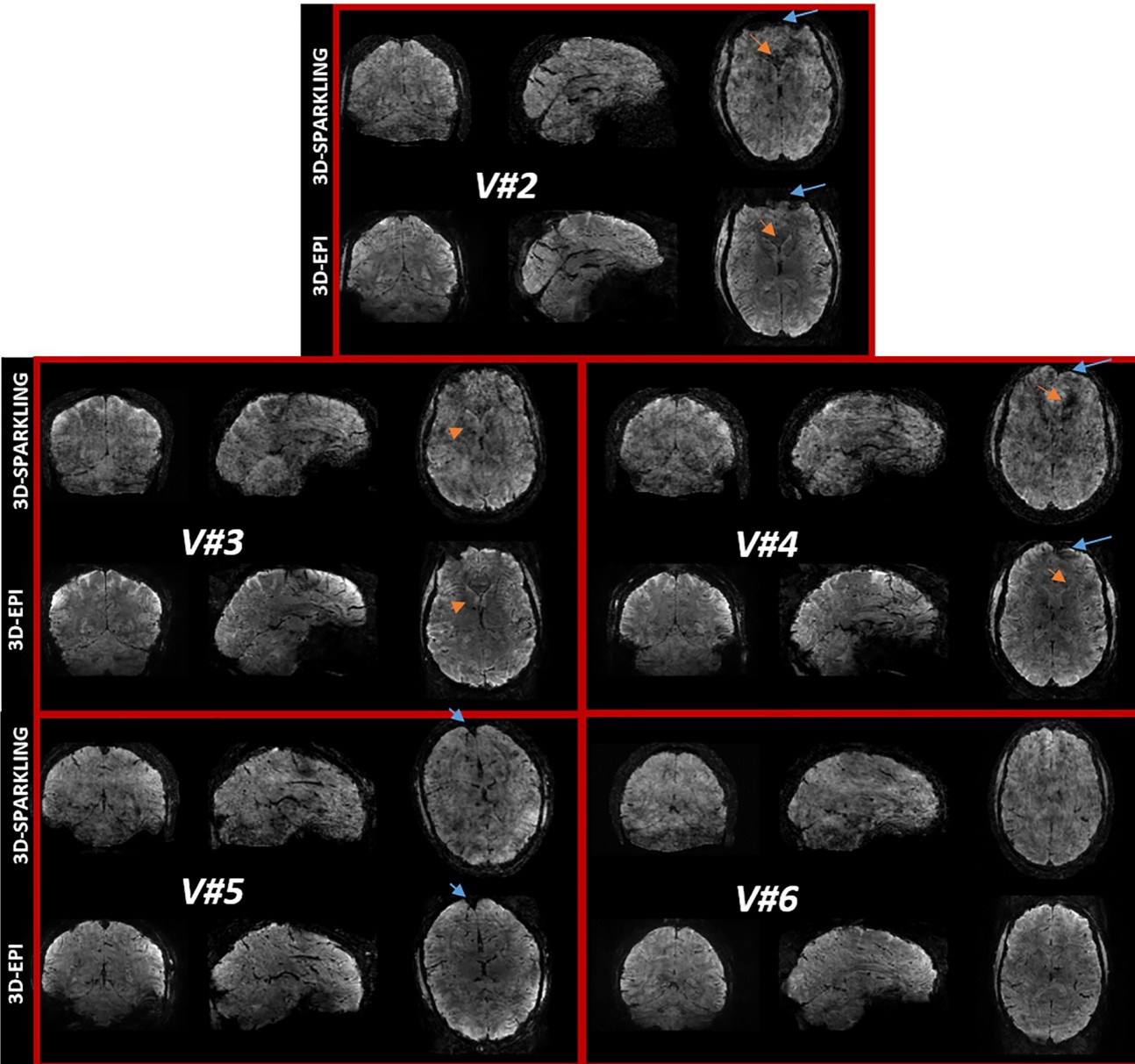

**Fig 3. Subject-wise comparison of the mean images derived from 3D-EPI and 3D-SPARKLING resting-state fMRI data.** Anatomical details that are lost in 3D-SPARKLING data are well recovered in 3D-EPI data. The overall contrast between tissues is clearer using the 3D-EPI encoding scheme. Signal loss and geometric distortions affect data collected using the two acquisition techniques differently. The blue and orange arrows point to brain areas where 3D-SPARKLING is respectively less and more affected by signal loss than 3D-EPI. The overall mean image quality of 3D-EPI data is superior to that of 3D-SPARKLING in all participants.

in five volunteers as previously mentioned. The tSNR values are higher on 3D-SPARKLING data, which suggests improved temporal stability and potentially a better sensitivity to the BOLD contrast, a feature that will be further analyzed hereafter. It is, however, important to keep in mind that a direct comparison between the tSNR yielded by the unbiased linear reconstruction applied to the k-space data collected by 3D-EPI and that yielded by the CS-based reconstruction applied to the k-space data collected by 3D-SPARKLING is limited as the latter induces some bias and therefore a reduced variance. However, as a moderate amount of $\ell_1$

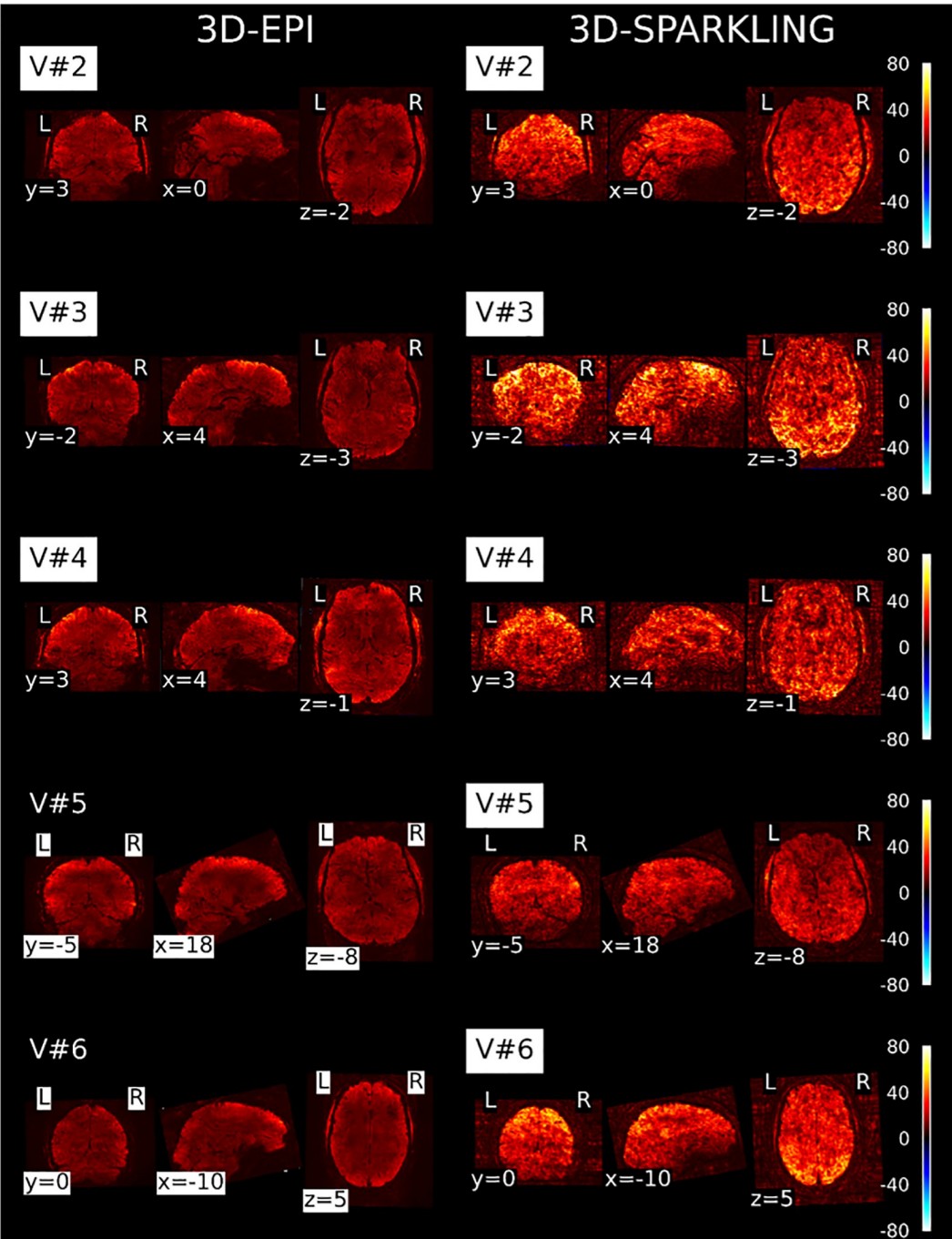

**Fig 4. Comparison of the tSNR maps derived from 3D-SPARKLING and 3D-EPI resting-state fMRI data.**
3D-SPARKLING data reveals a higher tSNR in comparison with 3D-EPI data. V#5 has a lower tSNR on average than the other participants, notably in the visual cortex and the posterior part of the brain.

regularization was used, the impact was limited. Additionally, we checked that the higher tSNR was primarily explained by the non-Cartesian encoding scheme and not by the nonlinear CS-based reconstruction. It is also worth mentioning that V#5 exhibits a lower tSNR on average than the other participants for both encoding schemes.

## Consistency of activation maps between the two encoding schemes

Fig 5 compares the significant global effects of interest estimated from the retinotopic fMRI data collected using 3D-EPI and 3D-SPARKLING. It shows that the activation patterns are well localized in the visual cortex (cf qCons1).

Compared to the other participants, V#5 elicits less activated voxels: $p < 0.001$ was too conservative to extract a significant activation pattern for both techniques. Due to this, we slightly relaxed the threshold to $\alpha = 0.05$ to gain insight on functional activity notably in 3D-SPARKLING. At this level of significance, the activation pattern associated with 3D-EPI data remains meaningless. This is likely a consequence of the lower tSNR observed in V#5, notably in the visual cortex as shown in Fig 4. Globally, the statistically significant activation patterns are relatively consistent across encoding schemes. This qualitative observation is supported by Table 2 which reports the F1-score (i.e. Dice index) values computed between the statistically significant activation patterns derived from 3D-SPARKLING and 3D-EPI retinotopic fMRI data (QCons1): As these values range between 0.82 and 0.99 across participants, this showcases that the spatial supports of these activation patterns are very consistent and cover the same brain areas. This simple yet meaningful sanity check between fMRI data collected using the two competing encoding schemes allowed us to look further into their potential differences as related to effect size i.e., height of activation peaks and recovery of retinotopic maps. Indeed, the activation maps in Fig 5 suggest that:

(i): 3D-SPARKLING outperforms 3D-EPI in V#1-V#3 and V#5.

(ii): 3D-EPI outperforms 3D-SPARKLING in V#4.

(iii): Both techniques perform similarly in V#6.

As 3D-SPARKLING (respectively, 3D-EPI) is run first for the volunteers indexed by odd numbers (respectively, even numbers, cf. Fig 1), these observations suggest that the differences in terms of effect size may not solely result from the difference in the sampling techniques but also be partly driven by the order of execution of the runs. In order to gain more insight into the impact of the encoding scheme versus the order of acquisition, the sensitivity to the BOLD effect was studied and compared between the two methods.

## Sensitivity to the BOLD effect elicited by task performance

Table 3 reports the number of activated voxels within the whole brain and in the gray matter for 3D-EPI and 3D-SPARKLING data (QSens1): The total number of voxels activated is larger in 3D-EPI data for five participants out of six, however, this number can be biased by false positives and it is far more problematic as no correction for multiple comparisons was performed. It is then more interesting to examine the number of activated voxels in the gray matter: The figures in Table 3 show that more voxels in the gray matter were systematically activated in the data collected first. This can be explained by a stimulus presentation effect also called the repetition suppression effect [76, 77].

Such an observation supports the hypothesis that the higher sensitivity to the BOLD effect is partly driven by the order of execution of the sequences, however, the number of activated voxels in gray matter only partially reflects the sensitivity to the BOLD effect. A more global insight can be obtained by examining the activation maps and their potential differences in terms of peak heights using both qualitative and quantitative metrics.

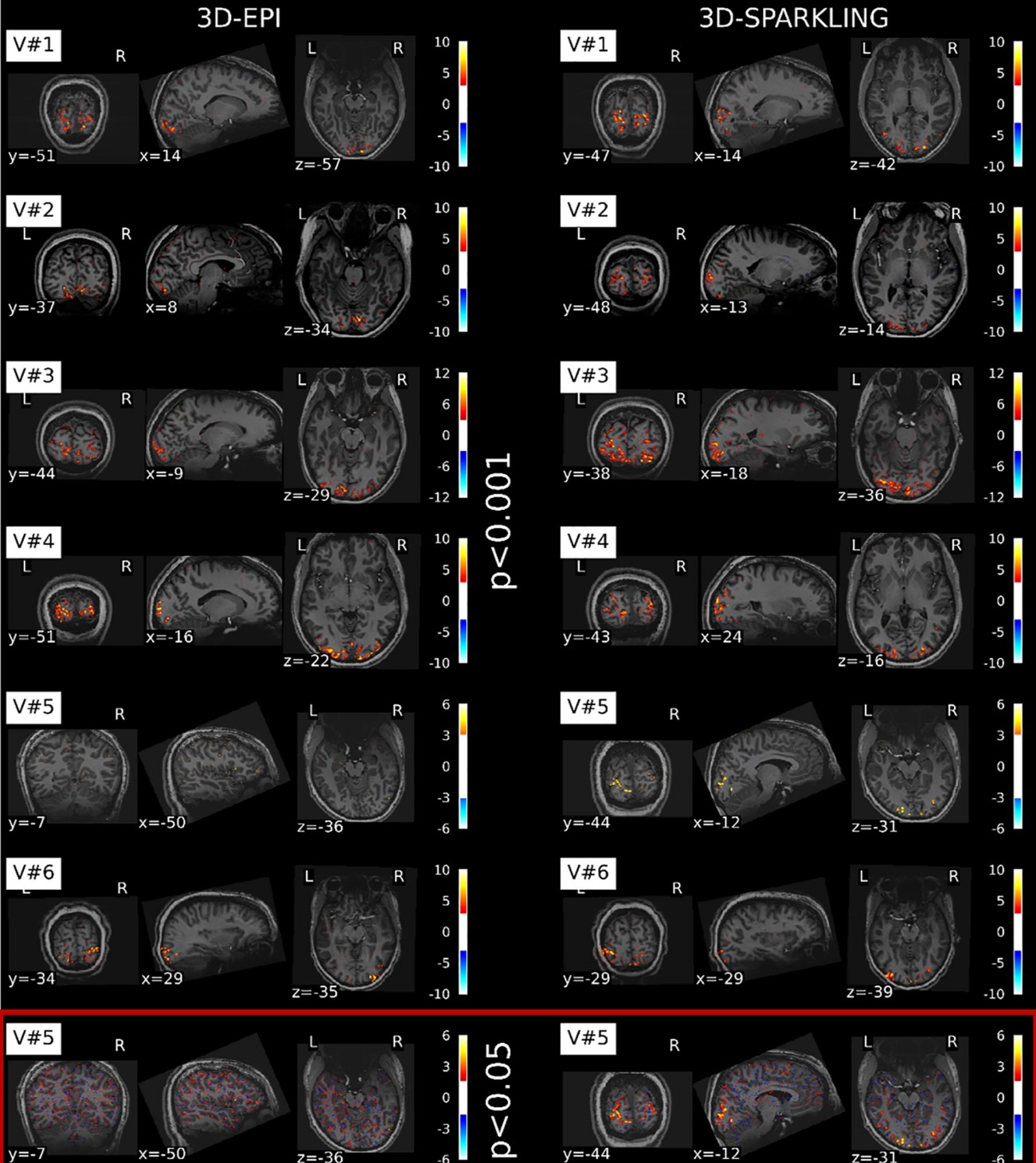

**Fig 5. Z-score maps derived from the global effects of interest.** From top to bottom: Activation maps displayed for the 6 participants and the two acquisition techniques. The six first rows show the activation maps yielded after thresholding the $z$-scores over the whole brain for $p < 0.001$ without applying any correction for multiple comparisons. The seventh row shows the activation map obtained after thresholding the $z$-scores over the whole brain for $p < 0.05$ with a minimum cluster size of 5 voxels for the fifth participant. Row-wise, the color bars are unchanged but differ from one volunteer to another. The slices were chosen according to the largest spatial extent of the activation patterns.

**Table 2. Comparison of the *F*1-scores computed between the activation patterns (thresholded *z*-score maps associated with the global effects of interest) derived from 3D-EPI and 3D-SPARKLING retinotopic fMRI data for the 6 volunteers.**

| Volunteer | F1-score |
|---|---|
| V#1 | 0.834 |
| V#2 | 0.925 |
| V#3 | 0.819 |
| V#4 | 0.820 |
| V#5 | 0.987 |
| V#6 | 0.933 |

These activation maps were produced by thresholding the *z*-scores over the whole brain for $p < 0.001$ (uncorrected for multiple comparisons). Despite the difference in significance, the activation patterns fit the gray matter quite well for both 3D-EPI and 3D-SPARKLING.

To do this, we show in Fig 6 the same axial slices of activation maps for volunteers V#3 and V#4 (qSens1): A higher statistical significance (or effect size) is observed for 3D-SPARKLING data in V#3 while the opposite statement holds in V#4 with more significant activations for 3D-EPI data. This confirms that the data collected first (i.e. 3D-SPARKLING in V#3, 3D-EPI in V#4) elicits more evoked brain activity.

Optimally from a statistical perspective, if we had a larger cohort, a two-way repeated measures analysis of variance (ANOVA) could have been performed voxelwise in order to disentangle the contribution of the order of sequence execution from that of the encoding scheme to the sensitivity to the BOLD effect. However, since we only collected fMRI data in six volunteers, we proceeded differently by first constructing for each subject the distribution of the statistically significant *z*-scores in the gray matter associated with 3D-EPI and 3D-SPARKLING retinotopic data, and then by testing the statistical difference between these two ensuing distributions using a Kolmogorov-Smirnov test (cf. QSens2). As we pulled the *z*-scores across voxels, this within-subject test is no longer spatially localized. However, it is a good indicator to use when deciding which encoding scheme elicits more evoked activity during task performance. Hence, in each individual we considered the following null hypothesis ($H_0$: $P_{EPI}(z) = P_{SPARK}(z), \forall z \geq 3.09$) and eventually rejected it for all participants (p-values ranging between $10^{-6}$ and $10^{-70}$) as shown in Table 4. In the latter we actually reported the unilateral (i.e. one-sided) p-values of the KS test we carried out subjectwise.

In five volunteers out of six, the distribution of activations elicited during 3D-SPARKLING retinotopic fMRI acquisitions is significantly shifted to the right (i.e. higher *z*-scores) compared to the similar distribution associated with 3D-EPI retinotopic fMRI data. The opposite

**Table 3. Comparison of the number of activated voxels (statistically significant at $p < 0.001$ without correcting for multiple comparisons) overall and in gray matter (GM) for 3D-EPI and 3D-SPARKLING data in the 6 volunteers.** The lowest numbers are retrieved in V#5.

| Volunteer | Total number of activated voxels | | Number of activated voxels in GM | |
|---|---|---|---|---|
| | **EPI** | **SPARK** | **EPI** | **SPARK** |
| V#1 | **7362** | 6661 | 4736 | **4863** |
| V#2 | **8423** | 5246 | **5528** | 3949 |
| V#3 | 9010 | **24410** | 6040 | **15263** |
| V#4 | **11779** | 4847 | **8350** | 3101 |
| V#5 | **3222** | 2606 | 1952 | **2265** |
| V#6 | **6437** | 4350 | **4198** | 2815 |

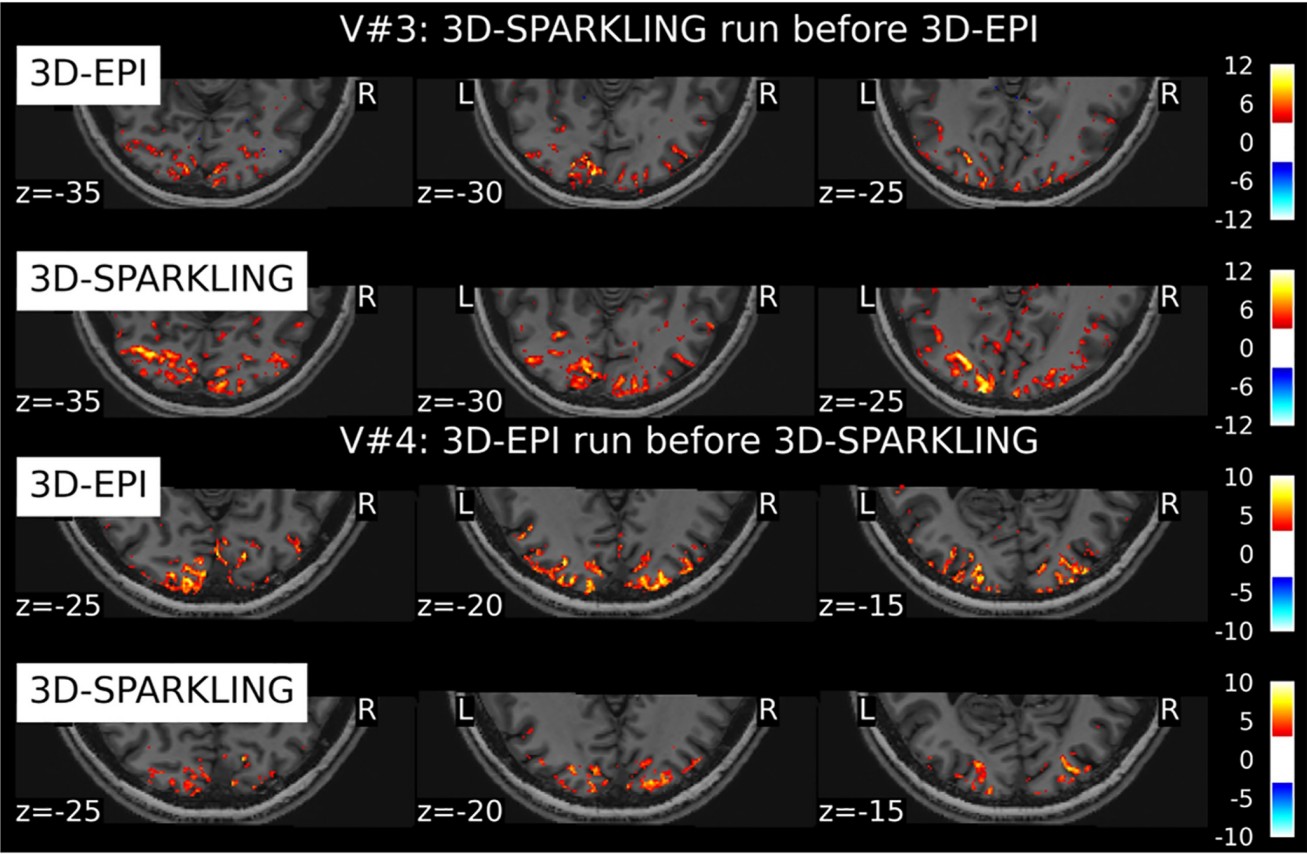

**Fig 6. Three axial slices showing the impact of the order of sequence execution on the statistical sensitivity and activation patterns similarity between the two encoding schemes in V#3 and V#4.** The sequence run first elicits more activation.

effect is only retrieved in V#4. Therefore, irrespective of the order of sequence execution, the 3D-SPARKLING encoding scheme and its associated preprocessing pipeline yield a higher statistical sensitivity to detect evoked brain activity in the gray matter compared to the 3D-EPI acquisition technique.

### Spatial specificity of activation maps on retinotopic fMRI data

In order to compare the two encoding methods according to the spatial specificity criterion `QSpe1`, the percentages of activated voxels within the GM, WM, CSF and other tissues were

**Table 4. Table summarizing the p-values and D-statistics of a Kolmogorov-Smirnov test between the distributions of the statistically significant $z$-scores within the gray matter extracted from 3D-EPI and 3D-SPARKLING retinotopic fMRI data:** We used the following null hypothesis $H_0$: $P_{EPI}(z) = P_{SPARK}(z)$, $\forall z \geq 3.09$ and separately in one-sided tests the two alternative hypotheses $H_{Alt}^{\text{Right}}$: $\exists z \geq 3.09 | P_{SPARK}(z) > P_{EPI}(z)$ and $H_{Alt}^{\text{Left}}$: $\exists z \geq 3.09 | P_{SPARK}(z) < P_{EPI}(z)$.

| Volunteer | $H_{Alt}^{\text{Right}}$ | | $H_{Alt}^{\text{Left}}$ | |
|---|---|---|---|---|
| | **D-statistic** | **p-value** | **D-statistic** | **p-value** |
| V#1 | 0.18 | $\mathbf{10^{-70}}$ | 0.0 | 1.0 |
| V#2 | 0.06 | $\mathbf{10^{-9}}$ | 0.005 | 0.84 |
| V#3 | 0.07 | $\mathbf{10^{-23}}$ | 0.001 | 0.98 |
| V#4 | 0.015 | 0.3489 | 0.053 | $\mathbf{10^{-6}}$ |
| V#5 | 0.29 | $\mathbf{10^{-79}}$ | 0.0004 | 0.99 |
| V#6 | 0.12 | $\mathbf{10^{-24}}$ | 0.0 | 1.0 |

**Table 5. Percentage of activated voxels in gray matter (%GM), white matter (%WM), cerebrospinal fluid (%CSF) and other tissues with regards to the total number of activated voxels for 3D-EPI and 3D-SPARKLING denoted respectively EPI and SPARK in each participant.** Significant p-values were thresholded at 0.001 uncorrected for multiple comparisons. The higher the better (in bold font) in the % GM column, the lower the better (in bold font) in others.

| Volunteer | %GM | | %WM | | %CSF | | %other tissues | |
|---|---|---|---|---|---|---|---|---|
| | EPI | SPARK | EPI | SPARK | EPI | SPARK | EPI | SPARK |
| V#1 | 64.33 | **73.01** | 20.71 | **18.75** | 11.06 | **8.05** | 3.9 | **0.19** |
| V#2 | 65.63 | **75.28** | **15.26** | 17.55 | 18.34 | **7.05** | 0.77 | **0.12** |
| V#3 | 60.04 | **62.53** | **16.27** | 26.49 | 16.13 | **10.84** | 7.56 | **0.14** |
| V#4 | **70.90** | 63.98 | **15.84** | 28.57 | 13.08 | **7.37** | 0.18 | **0.08** |
| V#5 | 60.58 | **86.57** | 30.94 | **10.09** | 8.27 | **3.34** | 0.25 | **0** |
| V#6 | **65.12** | 64.71 | **27.97** | 31.77 | 6.7 | **3.4** | 0.21 | **0.12** |
| Average | 64.43 | **71.01** | **21.17** | 22.2 | 12.26 | **6.68** | 2.15 | **0.1** |

extracted in each individual. Table 5 reports the main findings and shows that 3D-SPARKLING induces a higher percentage of activated voxels in the gray matter than 3D-EPI. The mean gap between the two encoding methods is quite large (71% for 3D-SPARKLING vs 64.5% for 3D-EPI). Similar analysis conducted in the white matter shows a marginal difference between 3D-SPARKLING (22% of activated voxels) and 3D-EPI (21%).

Although these average differences are quite small, they hide a large variability across individuals (V#2 vs V#5 for 3D-EPI, V#2 vs V#6 for 3D-SPARKLING) and even across encoding methods in the same participant (e.g. V#5).

Additionally, the percentage of activated voxels in the CSF and other tissues is on average twice (respectively, 20 times) higher for 3D-EPI than for 3D-SPARKLING, namely 12.27% (respectively, 2.15%) for 3D-EPI as compared to 6.68% (respectively, 0.11%) for 3D-SPARKLING.

These observations reveal first an increased average specificity of activations in gray matter using 3D-SPARKLING. This finding actually holds in 4 participants out of 6 including V#2 for whom 3D-EPI data were collected first.

Second, the fact that both encoding schemes yielded a fairly large mean percentage of activations in the white matter suggests a mismatch in the co-registration between the mean fMRI image and the anatomical scan on an individual basis. The higher mean percentage of activations in the white matter using 3D-SPARKLING data can be explained by the fact that the images are more blurry and the borders between white and gray matter less sharp.

Third, the larger amount of unexpected activations in the CSF and other tissues using 3D-EPI is questionable. As we demonstrated that this encoding scheme is associated with improved image quality, it is unlikely that this is due to an intrinsic lack of specificity. Instead, we suspect that the differences in addressing $B_0$ field inhomogeneities (TOPUP for 3D-EPI vs our own correction/reconstruction pipeline for 3D-SPARKLING) might induce co-registration mismatches, especially in the frontal cortex as pointed out by the yellow arrows in Fig 7. Such an explanation is consistent with the observations in Fig 3 where 3D-EPI is more impacted by signal loss in the frontal regions than 3D-SPARKLING. Of course, correcting the statistical tests for multiple comparisons might eradicate these false positives but in that case the statistical approach would fully hide the outlined intrinsic differences between the two encoding methods.

To further bring evidence that 3D-SPARKLING has intrinsically a better spatial specificity than 3D-EPI and that the higher mean percentage of activated voxels in gray matter is not merely a result of a mismatch in co-registration, Table 6 reports the number of activated voxels after thresholding the $z$-scores over the whole brain at a statistical threshold of $p < 0.05$ with FDR correction (QSpe2). The figures in Table 6 are in agreement with the numbers of

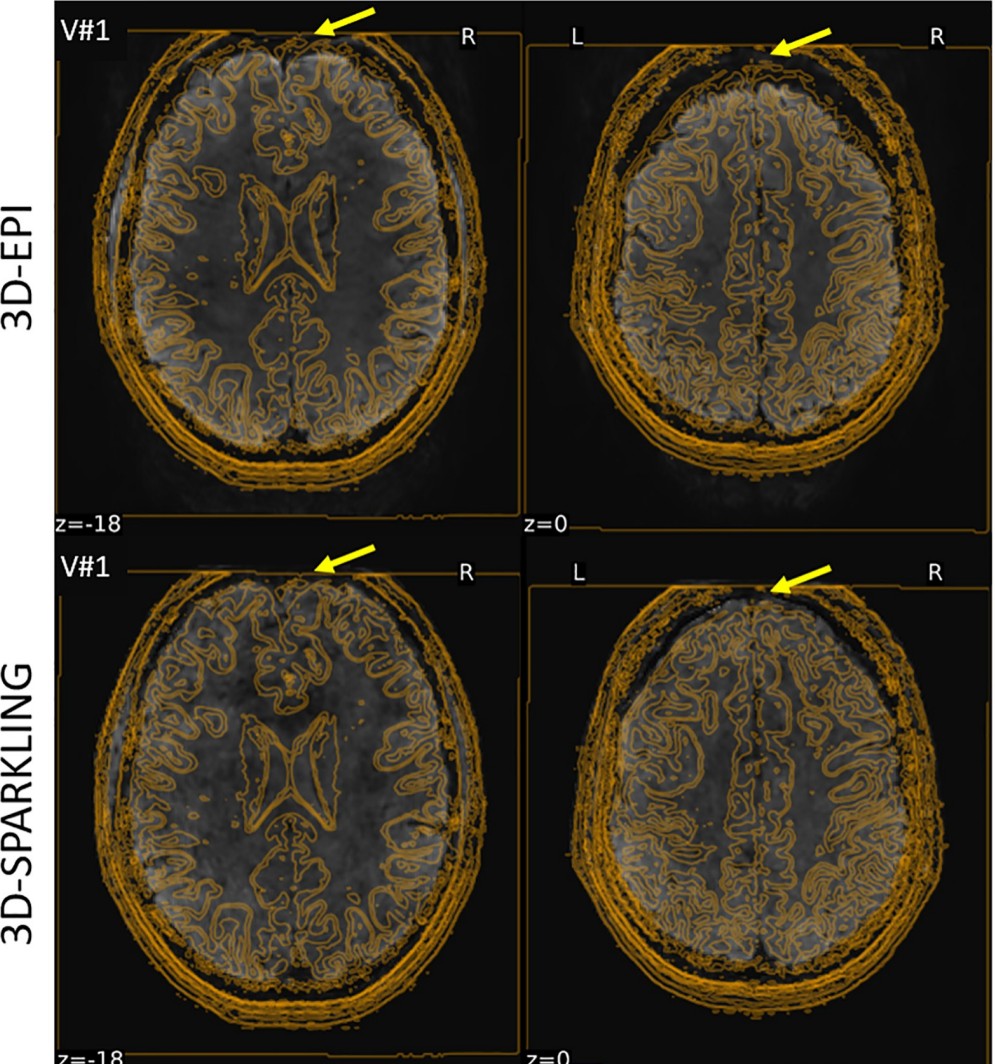

**Fig 7. The contour edges of the $T_1$-w anatomical scan overlaid on the mean fMRI images acquired using 3D-EPI and 3D-SPARKLING encoding schemes in V#1 during the retinotopic experiment.** The yellow arrows point to regions where the difference of mismatch between the functional and anatomical scans is visible.

**Table 6. Total number of activated voxels for 3D-EPI and 3D-SPARKLING data in the 6 volunteers.** These figures were retrieved by thresholding the $z$-scores over the whole brain at a p-value of 0.05 after FDR correction for multiple comparisons.

| Volunteer | Total number of activated voxels at p<0.05 after FDR correction | |
|---|---|---|
| | **3D-EPI** | **3D-SPARKLING** |
| V#1 | 3197 | **4535** |
| V#2 | **2853** | 1708 |
| V#3 | 4891 | **25193** |
| V#4 | **9392** | 3235 |
| V#5 | 4 | **660** |
| V#6 | **2722** | 2253 |

activated voxels in GM at $p < 0.001$ without correcting for multiple comparisons as originally reported in Table 3: In fact, we systematically found more activated voxels in the data collected first. This first demonstrates that the larger number of activated voxels yielded by 3D-EPI for five volunteers out of six when no correction for multiple comparisons is applied (cf. second column of Table 3) is merely biased by false positives. Second, it suggests that the data collected using 3D-SPARKLING yields functional maps that contain fewer false positives and more true positives at a given level of control in comparison with 3D-EPI, hence an improved spatial specificity.

### Accuracy of the retinotopic phase maps

To go one step further in examining the quality of the collected retinotopic fMRI data, we estimated the retinotopic phase maps (cf. Eq (9)) from 3D-SPARKLING and 3D-EPI raw fMRI data (i.e. spatially unsmoothed). In Fig 8 we visualize these maps on selected axial views to

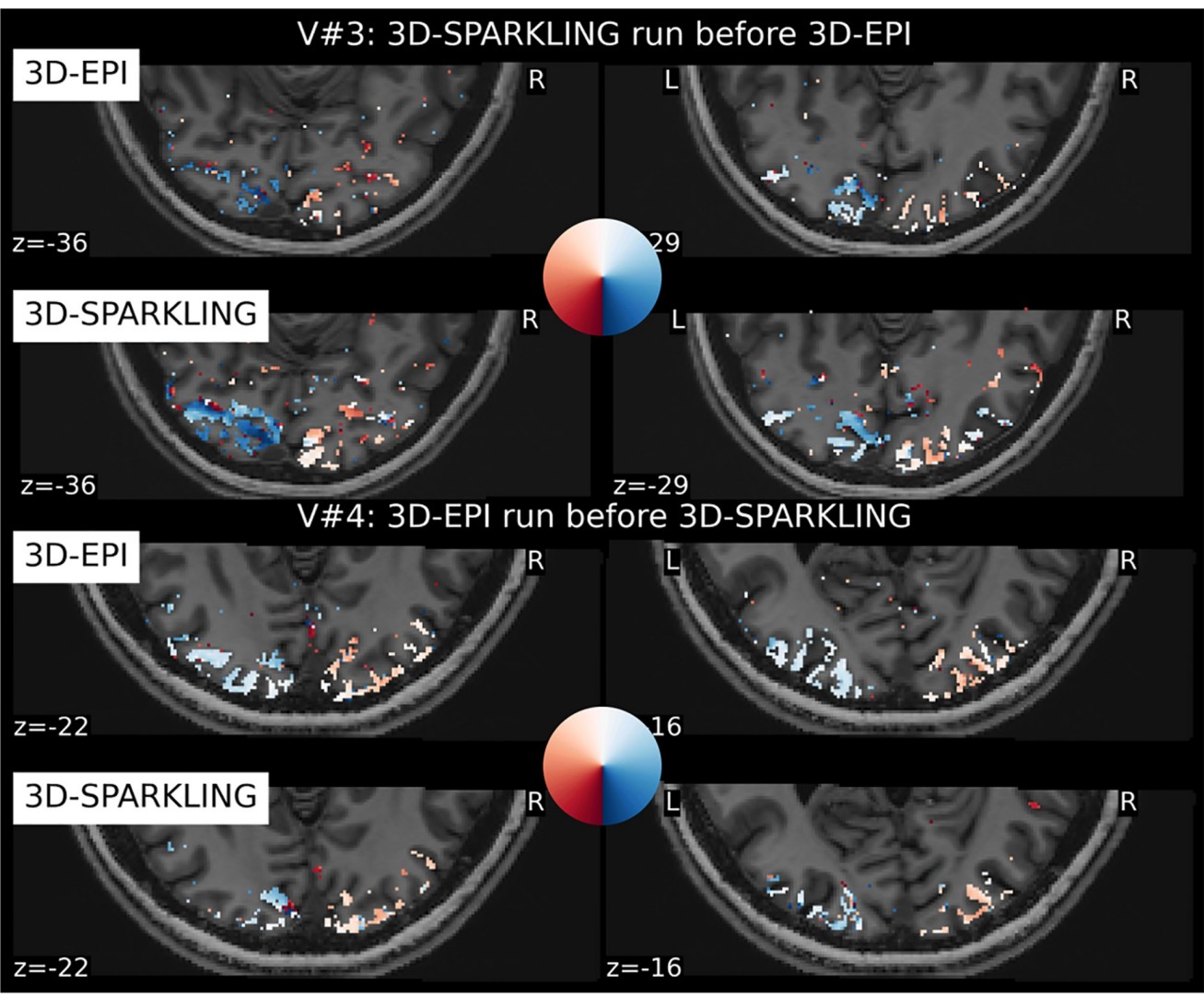

**Fig 8. BOLD phase maps computed for participants V#3 and V#4.** The BOLD phase maps agree with how the retina is supposed to be projected onto the visual cortex for both techniques.

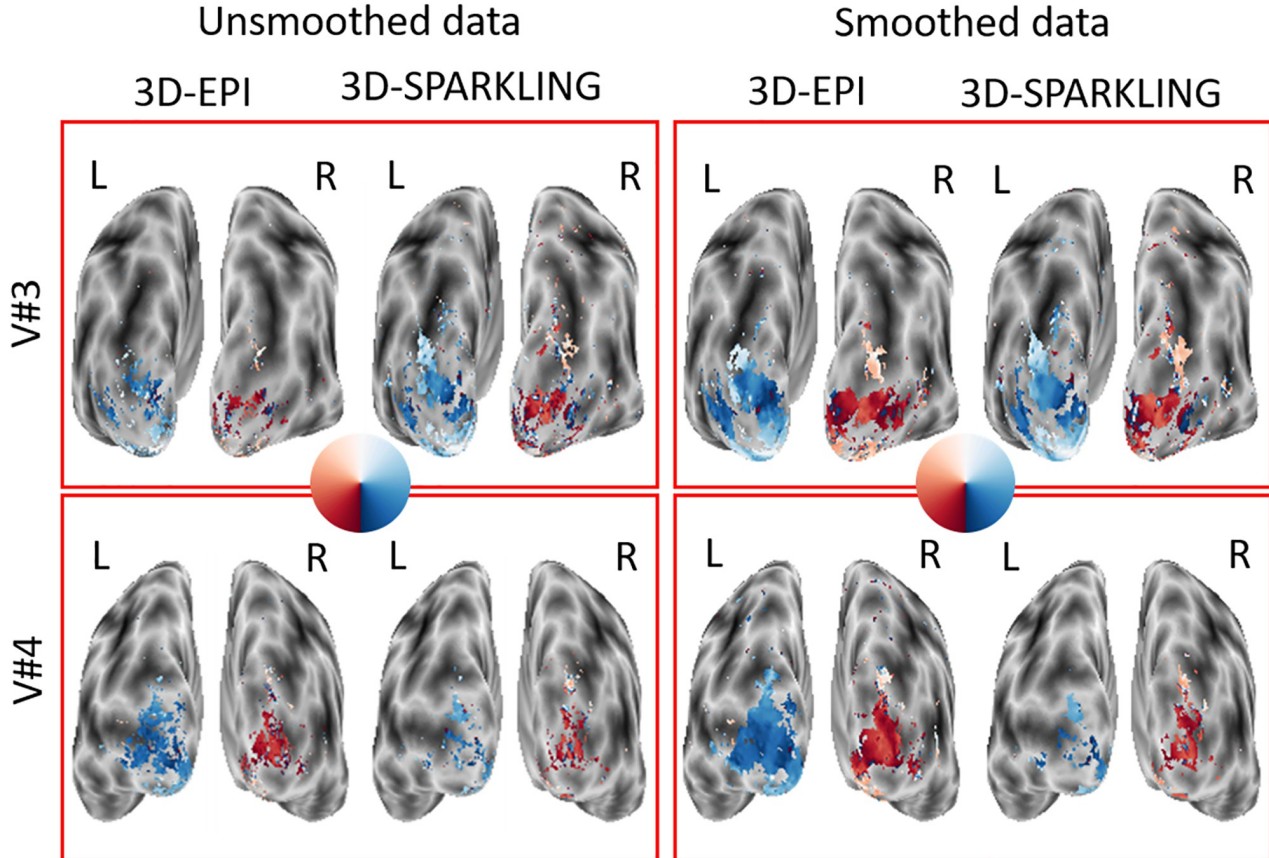

**Fig 9. Projection of the BOLD phase maps on the pial surface visualized on the inflated surface for participants V#3 (3D-SPARKLING run first) and V#4 (3D-EPI run first).** 3D-SPARKLING yields improved projected BOLD phase maps for V#3 in comparison with 3D-EPI both on raw and spatially smoothed data. Opposite results were found in favor of 3D-EPI in V#4, notably on spatially smoothed data.

check their consistency as well as their closeness to prior knowledge on the projection of the retina (represented by a colored disk in Fig 8) onto the visual cortex. As V#3 and V#4 were the most compliant volunteers during the whole fMRI experiment (no motion, results not shown), and as such, data reliability is enhanced and hence, a stronger evoked activity was elicited leading to more accurate retinotopic maps.

More precisely, 3D-SPARKLING data yields a higher quality phase map in V#3 (cf. top of Fig 8) with a smoother color coded (i.e. directional) gradient. In contrast, 3D-EPI data produced a more spatially extended and accurate phase map in V#4 (cf. bottom of Fig 8), reflecting again the order of sequence execution. As the visual fields in the retina are actually mirrored in the visual cortex, this means that the two visual hemifields project respectively onto the contra-lateral hemisphere in the visual cortex [78, 79]. For this particular reason, it is easier to visualize the projection of retinotopic maps on the cortical pial (qAccu1), as illustrated in Fig 9 for both raw (i.e. unsmoothed) and spatially smoothed (using a Gaussian kernel with a full width at half maximum (FWHM) of 2mm). fMRI data collected in V#3 and V#4. This 2-mm isotropic smoothing was applied during preprocessing prior to re-running a GLM analysis subjectwise.

The projected phase maps computed from the raw data illustrate once again the session effect as 3D-SPARKLING (resp., 3D-EPI) yields a larger effect for V#3 (resp., V#4). The data collected first yields a size effect that is large enough to retrieve an appreciable retinotopic

organization. In contrast, the effect size is smaller in the data collected second. It allows, how-ever, for a moderate recognition of the retinotopic organization. In addition to this, the results are overall consistent along the dorsal pathway: The blue (resp., red) color in the right (resp., left) hemifield projects to the left (resp., right) visual cortex.

The spatial smoothing at the preprocessing stage boosted the effect size and allowed us to retrieve more spatially extended phase maps, both for 3D-SPARKLING and 3D-EPI encoding schemes. Clearly, spatial smoothing was more beneficial to 3D-EPI than 3D-SPARKLING fMRI data : Smoothing has a greater impact on 3D-EPI data than on 3D-SPARKLING data in V#4. Nonetheless, this additional preprocessing step alters the intrinsic statistical sensitivity and specificity and is not entirely justified when only conducting within-subject statistical analyses.

## Discussion

This study seeks to evaluate the feasibility of 3D-SPARKLING as a non-Cartesian encoding scheme for high-resolution fMRI, specifically its sensitivity to the BOLD effect as well as its spatial specificity. The performance of 3D-SPARKLING was assessed on 6 volunteers and compared to the state-of-the-art Cartesian 3D-EPI encoding pattern at the same spatio-temporal resolution (1mm isotropic, 2.4s) at 7T. Image quality and tSNR metrics computed from resting-state data as well as outcomes from within-subject statistical analysis performed on retinotopic data were compared in each volunteer across the two acquisition methods.

### Main findings

First, regarding the resting-state fMRI data we show that 3D-EPI produces an improved image quality as compared to 3D-SPARKLING. On the other hand, 3D-SPARKLING yields higher tSNR in all participants over the whole brain but notably in the occipital lobe. On task-related fMRI data, across participants, we checked the validity of the retinotopic experiment, i.e. its ability to elicit evoked brain activity in the visual cortex at 7T for the set of acquisition parameters we considered, notably the spatio-temporal resolution and the two encoding schemes (qCons1 and QCons1). Despite the moderate quality of the BOLD phase maps yielded by the collected data, it is possible to retrieve a moderately reliable mapping of the visual areas on the cortical surface (qAccu1). Second, according to the criterion QSens2, we demonstrated that 3D-SPARKLING has an increased sensitivity to the BOLD effect in gray matter. Third, based on criteria QSpe1 and QSpe2, we proved that 3D-SPARKLING has an improved spatial specificity.

### How sensitivity to $\Delta B_0$ inhomogeneities in non-Cartesian fMRI impacts image quality and spatial specificity of detecting the BOLD effect?

The lower image quality observed on 3D-SPARKLING data in Fig 3 results from a higher sensitivity to static and dynamic $B_0$ inhomogeneities due to the random nature of the trajectories which led to the accumulation of differently oriented $\Delta B_0$ artifacts whereas, for 3D-EPI, the same readout is repeated across the k-space planes. Consequently, $B_0$ inhomogeneities in 3D-SPARKLING acquisitions yield severe blurring and signal loss, in contrast to the main geometric distortions in 3D-EPI images due to off-resonance effects. More generally, this sensitivity to $B_0$ imperfections is more prominent for some non-Cartesian encoding schemes. In regards to the dynamic fluctuations of the $B_0$ field, [80] showed on a T-Hex spiral-out encoding scheme that image quality was admissible in spiral imaging only when a full signal model was used for image reconstruction, i.e. when distortions related to both static and dynamic $B_0$ inhomogeneities were corrected. In this work, even though we tried to minimize the impact of

static $B_0$ inhomogeneities on 3D-SPARKLING data by correcting them retrospectively during image reconstruction, this correction was imperfect as a separately acquired $\Delta B_0$ field map was used for this purpose. Hence, any inconsistency between 3D-SPARKLING fMRI data and $\Delta B_0$ field map acquisitions (due to significant subject's motion for instance as in V#6) may significantly lower the impact of this correction. Moreover, we did not correct 3D-SPARKLING data from dynamic $B_0$ fluctuations, whereas a zeroth order correction was applied to 3D-EPI data.

As explained in Section "Image Quality and temporal SNR", this high sensitivity to $B_0$ imperfections affects the PSF of the acquisition and in turn can degrade the effective spatial resolution of 3D-SPARKLING data and consequently their spatial specificity. Nevertheless, this needs to be balanced with the PSF of the BOLD effect itself. In this respect, we noted that neither 3D-EPI nor 3D-SPARKLING have an actual spatial resolution of 1mm³. Beyond the sensitivity to $B_0$ fluctuations other sources of degradation such as the partial Fourier acceleration technique used in 3D-EPI or the strength of wavelet-based regularization implemented in the 3D-SPARKLING image reconstruction pipeline may play a significant role. Confronting 3D-SPARKLING's poor image quality observed in Fig 3 with its relatively good spatial specificity (64%-87% of activated voxels localized in gray matter), suggests that the PSF of the detected BOLD effect was not strongly affected. This could be partly explained by the fact that the spatial resolution we chose is slightly higher or very close to the theoretical BOLD PSF: In [81], using fMRI data collected at a $1 \times 1 \times 3$mm³ resolution, the authors showed that the BOLD PSF is less than 2mm in the primary visual cortex of the human brain. Later on, [82] used $0.9 \times 0.9 \times 1.0$mm³ fMRI data to estimate the BOLD PSF in the secondary visual cortex (V2) in the human brain. These authors reported that the BOLD PSF ranges between 0.83mm at the level of the WM/GM interface and 1.78mm at the border between the GM and CSF. As shown in Table 5, the relative percentage of activations retrieved in gray matter was higher in 3D-SPARKLING compared to 3D-EPI, and lower in the CSF and other tissues. This reflects a stronger spatial specificity or fewer false positives in 3D SPARKLING data. Concomitantly, the significant KS tests reported in Table 4 brought evidence that higher $z$-scores were obtained when using 3D-SPARKLING as encoding scheme suggesting a larger number of true positives.

Taken together, these results prove that despite lower image quality 3D-SPARKLING better preserves the temporal properties of the BOLD signal compared to 3D-EPI and may become even more competitive in the future with a full signal model including correction for dynamic $B_0$ inhomogeneities for image reconstruction.

## Challenges of high spatial resolution whole brain retinotopic mapping fMRI

In this work, we compared the activation and retinotopic maps derived from 3D-EPI and 3D-SPARKLING data: Our findings suggest that 3D-SPARKLING data has an improved sensitivity to the BOLD effect compared to 3D-EPI. This is likely due to the higher tSNR observed in such data which could be explained by the expected larger PSF and hence effective voxel size of 3D-SPARKLING data compared to 3D-EPI. A larger voxel size helps in aggregating more signals and therefore enhances the sensitivity at the cost of spatial specificity. We have demonstrated, however, that the spatial specificity of the data collected with 3D-SPARKLING is not severely impacted.

Furthermore, it is worth noticing that the reported unsmoothed retinotopic maps, whether obtained from 3D-EPI or 3D-SPARKLING data, are not as reliable as those usually reported in the literature. This is a direct consequence of the high resolution employed. In fact, the existing literature on retinotopic mapping is scarce concerning high resolution fMRI experiments as most studies are performed at a resolution around 3mm³ [83, 84].

Interestingly, in [85], the authors collected 1.1mm isotropic retinotopic fMRI data over 25 slices at 3T and 7T in a volumetric TR of 2s. Although this spatial resolution is close to ours, the retinotopic maps they obtained at 7T were more accurate and consistent with the state of the art than ours. It is a challenge to adequately explain this discrepancy as they used a significantly different experimental setup: 1) a different coil geometry, 2) a 2D imaging protocol with differing parameters and 3) a lower acceleration factor in parallel imaging. It is however interesting to note that they smoothed the projected phase maps using a Gaussian filter with a FWHM of 4mm. Applying the same strategy in Fig 9 but with a smaller FWHM of 2mm we recovered consistent and improved phase values. We marginally used this smoothing strategy as it is detrimental to the native spatial resolution. The low quality of the retinotopic maps retrieved from unsmoothed data tells us that this isotropic millimetric resolution remains challenging for a 10-minute high resolution whole brain fMRI retinotopy at 7T. However, given the variations of cortical thickness across lobes, choosing this resolution is a decent target to carefully analyse the spatial specificity of activation in gray matter.

Recent trends in high-resolution fMRI based either on 3D spiral encoding schemes [43] or on hybrid radial-EPI (TURBINE) k-space coverage [86] report enhanced evoked brain activity in the visual cortex as compared to ours obtained with both 3D-SPARKLING and 3D-EPI. Again, analyzing these differences is challenging as the authors employ a different set of parameters. Furthermore, we note that they rarely performed a direct comparison with the standard state-of-the-art (2D-SMS EPI or 3D-EPI) and/ or did not fully cover the brain. It is actually worth mentioning that when a high isotropic resolution is targeted, the 3D FOV very rarely covers the whole brain as otherwise higher acceleration factors are required which translates into a lower tSNR in 3D. Our results prove, however, that despite being challenging, our experimental setup is interesting and informative.

## Challenges related to comparing competing acquisition strategies in a few volunteers

In this study, we compared fMRI activation patterns from two data sets that were acquired using concurrent encoding schemes and image reconstruction strategies. Although we tried with due diligence to harmonize the experimental setup between both acquisition techniques by keeping very similar readout times, the same echo and repetition times, the same number of shots and very close fMRI preprocessing pipeline (except for $B_0$ distortion correction), we still encountered the limitations of comparing data sets that were acquired at different time points, notably due to the presence of confounding factors such as between-run subject's motion. Additionally, the different image reconstruction strategies used for 3D-EPI and 3D-SPARKLING data are entirely justified by the different acquisition techniques. On one hand, 2D CAIPIRINHA [36] was implemented within the 8-fold accelerated 3D-EPI sequence together with 6/8-fold partial Fourier. On the other hand, compressed sensing reconstruction with sparsity promoting prior in the wavelet domain [64] was used on 3D-SPARKLING data.

Following a linear reconstruction and in the thermal noise regime, the Gaussianity of the residual noise is generally judged as a realistic assumption. Such a claim may not be as straightforward following a nonlinear CS reconstruction. As the GLM framework supposes the Gaussianity of the residuals, it is then essential that both reconstructions do not bias the noise distribution. The Gaussianity assumption remains realistic in our case as well as illustrated by the experiment presented in the S1 Appendix.

With that in mind, Table 2 demonstrates that the shapes of the activation patterns elicited by the two encoding schemes are close to each other. Additionally, we paid attention to balancing the number of participants scanned first with 3D-EPI and 3D-SPARKLING acquisition

strategy to mitigate any potential loss in effect size due to the well know repetition suppression effect. We, therefore, think that despite some noticeable differences this comparative study remains valid and insightful.

That being said, we believe that the readout time, the unitary repetition time as well as the number of shots were mostly optimal for 3D-EPI and not for 3D-SPARKLING: Using either more shots by reducing the unitary TR or a longer readout time allowing us to collect more k-space data points can be implemented in 3D-SPARKLING and would be beneficial as it could improve image quality, higher tSNR and provides increased sensitivity and accuracy for retinotopic mapping. We however, decided not to implement such suggested changes either by using a shorter unitary TR with a larger number of shots or a longer readout time (which implies slightly mismatched TE values) for 3D-SPARKLING in order to avoid different $T_2^*$ contrasts between 3D-SPARKLING and 3D-EPI.

Hence for the sake of a fair comparison, we gave the priority to choosing the same acquisition parameters for the two encoding schemes over using the most efficient parameters for 3D-SPARKLING.

## Perspectives on how to further improve 3D-SPARKLING

It is worth noticing that similarly to 3D-EPI, both [43, 86] perform dynamic field fluctuation corrections in non-Cartesian imaging during image reconstruction. To do so, they used either external estimates from a field camera or their sequence in a self-navigating manner. Our next goal will be to investigate how this type of correction positively affects 3D-SPARKLING fMRI data. We have started to look at this aspect on dynamically monitored fMRI 3D-SPARKLING data based on a field camera. Our preliminary results showed consistent results with [43, 86], namely an improved image quality and higher tSNR maps. We also demonstrated an increased sensitivity to the BOLD contrast [87]. Moreover, similarly to TURBINE [86], 3D-SPARKLING is crossing the center of k-space at the center of each readout. This means that this encoding pattern can also be used in a self-navigating manner. Furthermore, this feature could be exploited to retrospectively reduce the volumetric TR and enhance the effective temporal resolution without sacrificing neither the spatial resolution nor the tSNR by implementing for instance a sliding window image reconstruction approach. In addition to this, analogously to TURBINE which implements temporal incoherence by using a golden angle increment between consecutive shots, 3D-SPARKLING could be extended to a 4D sampling pattern instead of being used in a simple scan-and-repeat mode. In that case, "low rank plus sparse" image reconstruction methods or local low-rank denoising techniques [88] could also be instrumental in obtaining high quality fMRI images at the native spatio-temporal resolution.

## Conclusion

In this work, 3D-SPARKLING was used for the first time to conduct whole brain fMRI acquisitions at 1mm$^3$ and for a temporal resolution of 2.4s. We also conducted an exhaustive comparison between 3D-SPARKLING and state-of-the-art 3D-EPI based on image quality, tSNR, sensitivity to the BOLD effect as well as spatial specificity following both qualitative and quantitative criteria. The results revealed that 3D-SPARKLING yields a higher tSNR, an improved sensitivity to the BOLD contrast and better spatial specificity than 3D-EPI. In addition to this, 3D-SPARKLING can be further enhanced by performing dynamic $B_0$ field fluctuations correction or exploiting the fact that each shot crosses the center of the k-space to expand it into 4D by implementing some incoherence in time between the shots.

## Supporting information

**S1 Appendix.**
(PDF)

## Acknowledgments

We thank Benedict A. Poser for providing us with the 3D-EPI sequence and the corresponding CAIPIRINHA reconstruction extension. We thank Alexis Amadon for his code for $\Delta B_0$ map estimation as well as Alexis Thual for his help debugging some features in `Nilearn`. We are grateful to Cecilia Garrec (CEA/DRF-Dir) for her comments, suggestions and editing of the manuscript with regards to the English language.

## Author Contributions

**Conceptualization:** Zaineb Amor, Alexandre Vignaud.

**Data curation:** Zaineb Amor.

**Formal analysis:** Philippe Ciuciu.

**Funding acquisition:** Philippe Ciuciu, Alexandre Vignaud.

**Investigation:** Zaineb Amor.

**Methodology:** Zaineb Amor, Philippe Ciuciu, Alexandre Vignaud.

**Project administration:** Alexandre Vignaud.

**Software:** Chaithya G. R., Guillaume Daval-Frérot, Franck Mauconduit, Bertrand Thirion.

**Supervision:** Philippe Ciuciu, Alexandre Vignaud.

**Validation:** Zaineb Amor, Philippe Ciuciu, Alexandre Vignaud.

**Visualization:** Zaineb Amor.

**Writing – original draft:** Zaineb Amor.

**Writing – review & editing:** Philippe Ciuciu, Chaithya G. R., Guillaume Daval-Frérot, Franck Mauconduit, Bertrand Thirion, Alexandre Vignaud.

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
