## [Decision Letter · Decision Letter 0]

29 May 2023

PONE-D-23-03044Non-Cartesian 3D-SPARKLING vs Cartesian 3D-EPI encoding schemes for functional Magnetic Resonance Imaging at 7 TeslaPLOS ONE

Dear Dr. Amor,

Thank you for submitting your manuscript to PLOS ONE. After careful consideration, we feel that it has merit but does not fully meet PLOS ONE’s publication criteria as it currently stands. Therefore, we invite you to submit a revised version of the manuscript that addresses the points raised during the review process.

We look forward to receiving your revised manuscript.

Kind regards,

Sumit Datta, PhD

Academic Editor

PLOS ONE

Journal Requirements:

"Chaithya G R was supported by the CEA NUMERICS program, which has received funding from the European Union’s Horizon 2020 research and innovation program under the Marie Sklodowska-Curie grant agreement No 800945. This work was granted access to the HPC resources of TGCC in France under the allocation 2019-GCH0424 made by GENCI. This work has received financial support from Leducq Foundation (large equipment Equipement de Recherche et Plateformes Technologiques program)."

"Chaithya G R was supported by the CEA NUMERICS program, which has received funding from the European Union’s Horizon 2020 research and innovation program under the Marie Sklodowska-Curie grant agreement No 800945. This work was granted access to the HPC resources of TGCC in France under the allocation 2019-GCH0424 made by GENCI. This work has received financial support from Leducq Foundation (large equipment Equipement de Recherche et Plateformes Technologiques program)."

7. We note that Figure 2 in your submission contain copyrighted image. All PLOS content is published under the Creative Commons Attribution License (CC BY 4.0), which means that the manuscript, images, and Supporting Information files will be freely available online, and any third party is permitted to access, download, copy, distribute, and use these materials in any way, even commercially, with proper attribution. For more information, see our copyright guidelines: http://journals.plos.org/plosone/s/licenses-and-copyright.

Additional Editor Comments:

We have received two expert reviews. However, both reviewers raise many critical issues. After carefully reading this work, I have to agree with the reviewers that the significance of this work is marginal. So, I recommendation a major revision of the your manuscript.

Reviewers' comments:

Reviewer's Responses to Questions

**Comments to the Author**

1. Is the manuscript technically sound, and do the data support the conclusions?

Reviewer #1: Partly

Reviewer #2: Yes

2. Has the statistical analysis been performed appropriately and rigorously? 

Reviewer #1: Yes

Reviewer #2: No

3. Have the authors made all data underlying the findings in their manuscript fully available?

Reviewer #1: Yes

Reviewer #2: No

4. Is the manuscript presented in an intelligible fashion and written in standard English?

Reviewer #1: No

Reviewer #2: Yes

5. Review Comments to the Author

Reviewer #1: The paper is generally well written and structured but I suggest minor re-.

The literature review was

thorough, It is comprehensive and

completely justifies

No technical errors and approved for publication

Reviewer #2: In this paper, a 3D-SPARKLING trajectory is evaluated for use in whole-brain isotropic functional MRI at 7 T. The proposed method uses a non-Cartesian readout, coupled with a non-linear reconstruction (compressed sensing with wavelet domain sparsity), and was compared to conventional 3D-EPI with matched parameters. While image quality in the 3D-SPARKLING acquisition was poorer than the 3D-EPI, most metrics showed improved sensitivity and specificty for the proposed method over 3D EPI, evaluated in 6 healthy volunteers.

The paper is reasonably well written, and many comprehensive experiments/evaluations are performed. However, I have several reservations about the results presented in the paper:

Major Comments:

1. The paper doesn't provide a figure or diagram of the resulting 3D SPARKLING trajectory. However, my understanding is that 3D SPARKLING optimized trajectories depend heavily on initialization, and in previous work, 3D SPARKLING trajectories were initialized with perturbed 3D radial trajectories. One would assume this to be the case here as well (although it is unstated as far as I can tell). Given this, it is quite confusing that the 3D-SPARKLING trajectories is stated to have been acquired in the "P-A encoding direction" (see Line 163). This is also stated in the bottom row of Table 1. It would be useful for the authors to explain in what way exactly the "P-A encoding direction" is meaningful in the context of 3D SPARKLING.

2. It is somewhat difficult to believe that despite poorer image quality, 3D-SPARKLING outperforms 3D-EPI in nearly every quantitative metric. As the BOLD signal is encoded in the image magnitude, one might expect that the blurring/aliasing seen in the image should also be reflected in the BOLD signals. The authors address this point in part by appealing to BOLD PSFs, but if it is expected that inclusion of dynamic off-resonance correction would significantly improve image quality, I believe the manuscript would significantly benefit from including that, rather than leaving it for future work. It would be difficult to recommend a method that produces qualitatively worse looking images, despite what the quantitative metrics show.

3. Another reasons CS has not been widely adopted in fMRI is due to the nonlinear reconstruction. In contrast with most linear reconstructions (e.g. parallel imaging), CS reconstructions are not unbiased, and furthermore, due to the non-linearity of the reconstruction, characteristics of the resulting "noise" is not straightforward. As most fMRI analysis tools have assumptions grounded in the Gaussianity of residuals, while this can be true for linear reconstructions (in the high SNR limit anyway when dealing with magnitude images), no guarantees about the distributional properties of the signal residuals can be made with CS reconstructions. This may be confounded further by interaction with image pre-processing. Therefore, naive adoption of fMRI analysis methods without consideration of altered noise characteristics is statistically unsound. At a minimum, some validation of the residual characteristics of the fMRI analysis models would help establish their statistical validity. Discussion of this important point should also be included in the manuscript. Although the reconstructions performed here are done independently volume-to-volume, if time-dependent regularization is performed, then additional considerations of any potential loss of temporal degrees of freedom are also necessary to ensure statistical validity of the inference process.

4. As a followup to point 3, evaluation of tSNR is not particularly meaningful when comparing biased (CS) vs. unbiased (PI) reconstructions. Between unbiased reconstructions, tSNR evaluation is appropriate as it is related to mean squared error (no bias). However, heavily regularized CS reconstructions could potentially have infinite tSNR, as variance is reduced to zero (as tSNR does not penalize bias). Therefore, some consideration of this fact should be taken into account when reporting tSNR.

6. PLOS authors have the option to publish the peer review history of their article (what does this mean?). If published, this will include your full peer review and any attached files.

Reviewer #1: No

Reviewer #2: No

---

## [Author Response · Author response to Decision Letter 0]

4 Sep 2023

We have enclosed/uploaded a PDF file entitled Response_to_Reviewers with our new Manuscript (marked and unmarked) where we respond to each point raised individually. We also replaced Figure 2 by a new one and uploaded a new Cover Letter with the information required about the funding.

---

## [Decision Letter · Decision Letter 1]

2 Oct 2023

PONE-D-23-03044R1Non-Cartesian 3D-SPARKLING vs Cartesian 3D-EPI encoding schemes for functional Magnetic Resonance Imaging at 7 TeslaPLOS ONE

Dear Dr. Amor,

Thank you for submitting your manuscript to PLOS ONE.   As suggested by the reviewer, authors are requested to recheck and correct the typographical or grammatical mistakes if any. Therefore, we invite you to submit a revised version of the manuscript.

We look forward to receiving your revised manuscript.

Kind regards,

Sumit Datta, PhD

Academic Editor

PLOS ONE

Journal Requirements:

Additional Editor Comments:

Authors address all the queries resaid by reviewers. As suggested by the reviewer, authors are requested to recheck and correct the typographical or grammatical mistakes if any.

Reviewers' comments:

Reviewer's Responses to Questions

**Comments to the Author**

1. If the authors have adequately addressed your comments raised in a previous round of review and you feel that this manuscript is now acceptable for publication, you may indicate that here to bypass the “Comments to the Author” section, enter your conflict of interest statement in the “Confidential to Editor” section, and submit your "Accept" recommendation.

Reviewer #1: All comments have been addressed

Reviewer #2: All comments have been addressed

2. Is the manuscript technically sound, and do the data support the conclusions?

Reviewer #1: Yes

Reviewer #2: Yes

3. Has the statistical analysis been performed appropriately and rigorously? 

Reviewer #1: (No Response)

Reviewer #2: Yes

4. Have the authors made all data underlying the findings in their manuscript fully available?

Reviewer #1: Yes

Reviewer #2: No

5. Is the manuscript presented in an intelligible fashion and written in standard English?

Reviewer #1: No

Reviewer #2: Yes

6. Review Comments to the Author

Reviewer #1: I require authors to make some minor revisions

article must be clear and correct, some typographical errors must be corrected.

Also some grammatical errors should be corrected.

Reviewer #2: (No Response)

7. PLOS authors have the option to publish the peer review history of their article (what does this mean?). If published, this will include your full peer review and any attached files.

Reviewer #1: No

Reviewer #2: No

---

## [Author Response · Author response to Decision Letter 1]

22 Oct 2023

We have enclosed our response in a separate file entitled Response_to_Reviwers.

---

## [Decision Letter · Decision Letter 2]

19 Feb 2024

Non-Cartesian 3D-SPARKLING vs Cartesian 3D-EPI encoding schemes for functional Magnetic Resonance Imaging at 7 Tesla

PONE-D-23-03044R2

Dear Dr. Amor,

We’re pleased to inform you that your manuscript has been judged scientifically suitable for publication and will be formally accepted for publication once it meets all outstanding technical requirements.

Kind regards,

Ozlem Ipek, Ph.D.

Academic Editor

PLOS ONE

Additional Editor Comments (optional):

Reviewers' comments:

Reviewer's Responses to Questions

**Comments to the Author**

1. If the authors have adequately addressed your comments raised in a previous round of review and you feel that this manuscript is now acceptable for publication, you may indicate that here to bypass the “Comments to the Author” section, enter your conflict of interest statement in the “Confidential to Editor” section, and submit your "Accept" recommendation.

Reviewer #1: (No Response)

Reviewer #2: All comments have been addressed

2. Is the manuscript technically sound, and do the data support the conclusions?

Reviewer #1: Yes

Reviewer #2: (No Response)

3. Has the statistical analysis been performed appropriately and rigorously? 

Reviewer #1: Yes

Reviewer #2: (No Response)

4. Have the authors made all data underlying the findings in their manuscript fully available?

Reviewer #1: Yes

Reviewer #2: (No Response)

5. Is the manuscript presented in an intelligible fashion and written in standard English?

Reviewer #1: Yes

Reviewer #2: (No Response)

6. Review Comments to the Author

Reviewer #1: (No Response)

Reviewer #2: (No Response)

7. PLOS authors have the option to publish the peer review history of their article (what does this mean?). If published, this will include your full peer review and any attached files.

Reviewer #1: No

Reviewer #2: No

---

## [Editor Report · Acceptance letter]

29 Apr 2024

PONE-D-23-03044R2 

PLOS ONE

Dear Dr. Amor, 

I'm pleased to inform you that your manuscript has been deemed suitable for publication in PLOS ONE. Congratulations! Your manuscript is now being handed over to our production team.

Kind regards, 

on behalf of

Dr. Ozlem Ipek 

Academic Editor

PLOS ONE